# Mechanistic Interpretability as Statistical Estimation: A Variance Analysis

**Maxime Méloux** [1]   **François Portet** [1]   **Maxime Peyrard** [1]

## Abstract

Mechanistic Interpretability (MI) aims to reverse-engineer model behaviors by identifying functional sub-networks. Yet, the scientific validity of these findings depends on their stability. In this work, we argue that circuit discovery is not a standalone task but a statistical estimation problem built upon causal mediation analysis (CMA). We uncover a fundamental instability at this base layer: exact, single-input CMA scores exhibit high intrinsic variability, implying that the causal effect of a component is a volatile random variable rather than a fixed property. We then demonstrate that circuit discovery pipelines inherit this variability and further amplify it. Fast approximation methods, such as Edge Attribution Patching and its successors, introduce additional estimation noise, while aggregating these noisy scores over datasets leads to fragile structural estimates. Consequently, small perturbations in input data or hyperparameters yield vastly different circuits. We systematically decompose these sources of instability and advocate for more rigorous MI practices, prioritizing statistical robustness and routine reporting of stability metrics.

## 1. Introduction

As AI systems are increasingly deployed in real-world applications, the need for robust interpretability methods has become more urgent. Understanding the internal mechanisms of these models is critical not only for diagnosing failures and improving robustness (Barredo Arrieta et al., 2020), but also for complying with emerging legal frameworks that mandate explainability (Walke et al., 2025).

Mechanistic Interpretability (MI) is a promising research

direction aiming to reverse-engineer the algorithms learned by deep neural networks (Olah et al., 2018). A central approach in MI involves identifying "circuits", functional sub-networks that are responsible for particular capabilities (Olah et al., 2020; Elhage et al., 2021). These are typically identified by relying on the framework of causal mediation analyses (CMA) (Pearl, 2001; VanderWeele, 2016). CMA consists of intervening on the computational graph, setting the network in counterfactual states and measuring the effect of components on outputs (Vig et al., 2020; Monea et al., 2024; Hanna et al., 2024; Syed et al., 2024). In practice, MI relies on fast approximation of CMA to scale the estimation of causal importance scores to larger models, e.g., attribution patching (EAP; Syed et al., 2023) with integrated gradients (EAP-IG; Hanna et al., 2024). The causal importance scores are then aggregated over a dataset of inputs representative of the target behavior and discrete heuristics are applied to extract a *causally important* circuit. The long-term vision of MI is to evolve into a rigorous science, employing discovery tools similar to those of the natural sciences (Cammarata et al., 2020; Lindsey et al., 2025).

However, MI currently faces foundational challenges that limit its scientific rigor. Methods are prone to "dead salmon" artifacts and false positives (Méloux et al., 2025), and explanations discovered in one setting may fail to transfer to others (Hoelscher-Obermaier et al., 2023). In addition, multiple incompatible explanations may equally satisfy current MI criteria (Méloux et al., 2025). Méloux et al. (2025) argues that these issues stem from **non-identifiability**: the impossibility of inferring a unique explanation from observed data. In statistics, this non-identifiability manifests as high variance (Preston et al., 2025; Arendt et al., 2012).

These issues call for applying standard tools of statistical inference to MI (Fisher, 1955; Mayo, 1998), consistent with the principles of veridical science (Yu & Kumbier, 2020). In the natural sciences, validity requires quantifying observational variability and representing uncertainty (Lele, 2020; Committee et al., 2018). Systematically studying the stability of MI findings through metrics like variance (Zidek & van Eeden, 2003) is a necessary step toward scientific rigor. Yet, current MI practices often neglect these requirements; explanations are frequently reported without quantifying their statistical stability, robustness to perturbations, and uncertainty estimates (Rauker et al., 2023). Without such

[1]Université Grenoble Alpes, CNRS, Grenoble INP, LIG, 38000 Grenoble, France. Correspondence to: Maxime Méloux <maxime.meloux@univ-grenoble-alpes.fr>, Maxime Peyrard <maxime.peyrard@univ-grenoble-alpes.fr>.

*Proceedings of the 43rd International Conference on Machine Learning*, Seoul, South Korea. PMLR 306, 2026. Copyright 2026 by the author(s).

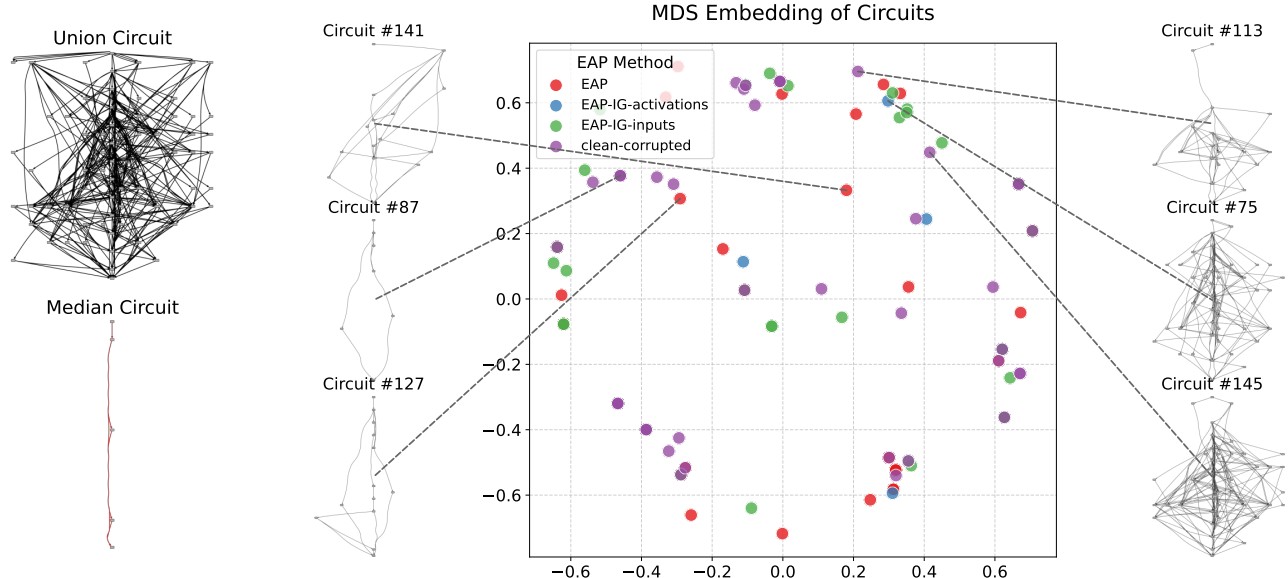

*Figure 1.* In gpt2-small, varying multiple circuit-finding parameters at once (type of resampling, aggregation, intervention, estimation, and pruning) yields many different circuits for the Greater-Than task, displayed along with the union and median circuit (left). The MDS projection of the pairwise Jaccard index matrix (center) shows that no method consistently yields lower variability (clustered) circuits. The detailed overlap between circuits, as well as a stability analysis, are provided in Appendix 6.4.

analyses, we cannot assess the generalizability, reliability, and ultimately, the validity of MI explanations (Rauker et al., 2023; Liu et al., 2025; Ioannidis, 2005).

At the heart of importance estimation in MI lies **causal mediation analysis (CMA)** (Pearl, 2001; VanderWeele, 2016). CMA provides a theoretical framework to estimate the causal effect of specific model edges or nodes on a behavior by mediating information through them. While CMA is identifiable at the level of a single input and behavior, the broader goal of circuit discovery is to aggregate these individual importance scores into a sparse, generalizable subgraph. In this work, we argue that circuit discovery should be viewed as a downstream pipeline fueled by CMA.

In this work, we analyze the stability of causal mediation analysis (CMA), its approximations (EAP, EAP-IG), and the downstream circuits they extract. We consider two sources of variability: (i) **data-related factors**, such as datasets (via bootstrap resampling), shifts in the distribution, prompt paraphrasing, and the choice of contrastive perturbation, and (ii) **methodological factors** such as hyperparameters and heuristics. We find substantial instability at every stage: CMA exhibits high variability in estimating causal importance across inputs drawn from the same distribution; its approximations further amplify this instability; and all circuit extraction methods produce highly unstable circuits across nearly all sources of variability. This instability is summarized in Fig. 1, which shows the structural inconsistency among circuits discovered when multiple parameters are varied simultaneously. In response, we propose a set of

best practices for the MI community, including systematic bootstrap resampling and reporting stability metrics, to promote more rigorous and reliable interpretability research.

## 2. Related Work

**Causal Mediation Analysis as the Engine of MI.** Causal mediation analysis (CMA; Pearl, 2001; VanderWeele, 2016) investigates how an outcome (e.g., a model's prediction) is affected by specific mediators (neuron activations or edges) via controlled interventions. In deep neural networks, this involves techniques such as activation patching (Vig et al., 2020; Geiger et al., 2021) and causal tracing (Meng et al., 2022; 2023; Fang et al., 2025), which manipulate mediators to quantify their influence on restoring a partially corrupted input. The causal effect of a component is identifiable for a fixed input and a fixed input corruption and can be computed exactly by simulating the execution of the networks under different interventions (Vig et al., 2020; Meng et al., 2022). However, exact CMA is computationally expensive, it involves several forward passes to estimate the causal effect of a single component. Consequently, the field has developed fast approximations. Edge Attribution Patching (EAP; Syed et al., 2023) combine causal patching with local Taylor expansion to quantify the importance of individual edges. EAP with integrated gradients (EAP-IG; Hanna et al., 2024) builds on this by using path integrals to better handle non-linearities and measures the impact of components excluded from a subgraph. One prominent application of these importance estimates is **circuit discovery**: a structural

estimation problem where one seeks to identify a sparse, interconnected subgraph (circuit) consisting of causally important components. This process has evolved from early techniques such as feature visualization (Zeiler & Fergus, 2014; Sundararajan et al., 2017) to automated methods such as ACDC (Conmy et al., 2023). Going from an estimated causal importance score for each component of a network to a discrete sub-graph selection involves several heuristics and design choices, leading to different algorithms.

**The Limits of Point-Estimate Evaluation.** Despite their grounding in causal theory, these methods produce **point estimates**: single structural summaries derived from finite data and fixed hyperparameters. Yet the notion of a unique, correct circuit is often ill-defined or non-identifiable (Mueller et al., 2026; Méloux et al., 2025), undermining claims about recovering a "ground-truth" circuit. More broadly, Méloux et al. (2025) argues for reframing interpretability as a problem of statistical explanation. Under this view, circuits should be reported with uncertainty estimates, since multiple distinct circuits may plausibly explain the same behavior. This shifts attention to variance: *how different are the circuits that are consistent with the evidence?*

Currently, MI relies on proxy metrics to evaluate those estimates based on desirable properties: **faithfulness** (how accurately a circuit reflects model behavior, often tested by perturbing or ablating the identified components within the full model; Conmy et al., 2023; Hedström et al., 2023; Hanna et al., 2024; Shi et al., 2024b), **sufficiency** (whether the isolated circuit can reproduce the target behavior; Bau et al., 2017; Yu et al., 2025; Shi et al., 2024a), **interpretability** (a qualitative assessment of understandability and alignment with intuition; Olah et al., 2020), and **sparsity/minimality** (a preference for simpler, concise circuits; Elhage et al., 2021; Hedström et al., 2023; Dunefsky et al., 2024; Shi et al., 2024a). While these assess the *internal validity* of a discovered circuit, they do not account for its *stability*.

Recent work has begun to question the robustness of these metrics. For instance, Shi et al. (2024a) introduce hypothesis tests for faithfulness, but only for a fixed circuit, while our work focuses on the stability of both circuits and CMA. While bootstrapping has been used to improve the selection of faithful edges (Nikankin et al., 2025), our study provides the first systematic decomposition of these instabilities. We trace the sources of instability across the pipeline: the baseline variance of single-input CMA, the approximation noise introduced by attribution methods, and the sensitivity to methodological choices. This mirrors the shift in classic ML from simple error rates to the study of model stability and generalization variance (Bousquet & Elisseeff, 2002).

**Identifying the Sources of Instability.** A growing body of evidence suggests that MI methods suffer from soundness issues. Interventions based on discovered circuits often fail to generalize to novel contexts, casting doubts on the robustness of the underlying identified mechanism (Hoelscher-Obermaier et al., 2023). Furthermore, results can be sensitive to the choice of perturbation strategies (Miller et al., 2024; Bhaskar et al., 2024; Zhang & Nanda, 2024).

These issues can be symptoms of **non-identifiability**: multiple distinct and incompatible circuits can equally satisfy common evaluation metrics (Méloux et al., 2025). This statistically appears as high estimator variance (Preston et al., 2025) and estimate instability due to the high dimensionality of the model and the limitations of finite sampling, stressing the need for proper uncertainty and stability analyses.

## 3. Formal Background

We present a brief formal description of CMA and underlying circuit discovery. For details, we point the reader to (Mueller et al., 2026). We highlight the statistical perspective on CMA and circuit discovery (Méloux et al., 2025).

### 3.1. Causal Mediation Analysis

The theoretical framework for identifying functional components in neural networks is causal mediation analysis (Pearl, 2001; VanderWeele, 2016). CMA investigates how an antecedent $X$ (input) affects an outcome $Y$ (output) through a mediator $M$ (an internal component such as a node or edge), partitioning the Total Effect (TE) of the input into direct and indirect pathways. In the context of MI, we focus on the **natural indirect effect (NIE)**: the portion of the effect that is transmitted specifically through the mediator (Mueller et al., 2026). Formally, let $Y(x, m)$ denote the value of the model $f_\theta$'s output metric $\mathcal{L}$ (e.g., logit difference or loss) under two distinct interventions (setting the input to $X = x$ and fixing the mediator to $M = m$). Standard activation patching techniques (Geiger et al., 2021; Vig et al., 2020) estimate this effect by contrasting two conditions: a clean run with input $x$, and a counterfactual run where the mediator is set to the value it would take under a corrupted input $x_{\text{corr}}$. The importance score $S$ for a component $e$ is defined as the NIE of transitioning the mediator from its clean to its corrupted state[1] in the context of the clean input:

$$S(e, x, x_{\text{corr}}) = \underbrace{\mathbb{E}[Y(x, M_e(x_{\text{corr}}))]}_{\text{Patched run}} - \underbrace{\mathbb{E}[Y(x, M_e(x))]}_{\text{Clean run}}$$

(1)

Here, $M_e(x)$ and $M_e(x_{\text{corr}})$ represent the natural value of the mediator in the clean and corrupted settings, respectively.

---

[1] Prior studies such as ACDC, EAP, and EAP-IG commonly consider the opposite effect of activation *restoration* instead (from the corrupted to clean state). Here, we keep the original formulation of CMA (Pearl, 2001; Vig et al., 2020; Mueller et al., 2026).

The importance score depends on two interventions: one setting the global context by transforming $x$ into $x_{\text{corr}}$ and one manipulating the mediator from $M_e(x)$ to $M_e(x_{\text{corr}})$.

Consequently, the estimated importance is not a fixed property of the component, but a random variable that depends on the joint distribution of the clean input $x$ and the counterfactual source $x_{\text{corr}}$. Fluctuations in how $x$ is sampled or how $x_{\text{corr}}$ is generated directly introduce variance into the definition of the score itself.

### 3.2. Circuit Discovery as Statistical Estimation

The following formulation uses standard concepts from statistical estimation, which we make explicit here. Despite their simplicity, these concepts are rarely applied in MI practice: circuit discovery results are seldom reported with uncertainty estimates or stability analyses.

While CMA provides precise per-input explanations for a specific input-counterfactual pair, MI typically seeks **global** (population-level) circuits: subgraphs that explain model behavior across a distribution representing a behavior of interest. Circuit discovery can be seen as a statistical estimation problem that generalizes these per-input CMA scores to a population-level circuit.

**Target Parameter:** Circuit discovery methods implicitly assume the existence of a population-level importance score for each component $e$. We define this target $\mu_e$ as the expected value of the per-input NIE scores over the joint distribution $\mathcal{D}$ of inputs $X$ and experimental conditions: $\mu_e = \mathbb{E}_{(x, x_{\text{corr}}) \sim \mathcal{D}}[S(e, x, x_{\text{corr}})]$. Since the full distribution $\mathcal{D}$ is inaccessible, methods rely on a finite dataset $D = \{(x_i, x_{\text{corr},i})\}_i$ sampled from $\mathcal{D}$ to estimate $\mu_e$ as the empirical mean (other aggregation methods could be used).

**Circuit Selection:** The final circuit $C$ is obtained by applying a selection procedure $\mathcal{A}$ to the set of aggregated scores, subject to hyperparameters $\Lambda$ such as sparsity thresholds or connectivity constraints. Formally, $C = \mathcal{A}(\{\hat{S}(e)\}_{e \in f_\theta}, \Lambda)$ where $f_\theta$ denotes the full model's computational graph.

This formulation highlights that a circuit is not solely a product of the model, but a compound effect of the estimation pipeline. The importance score $S$ exhibits intrinsic variance due to the sampling of inputs and perturbations. Also, the pipeline depends on the choice of hyperparameters and the selection function $\mathcal{A}$ can amplify small fluctuations in $\hat{S}(e)$ into large structural differences in $C$.

### 3.3. Approximating CMA via the EAP family

Calculating the exact NIE (Eq. 1) for every edge is computationally prohibitive ($2 \times N_{edges} \times N_{samples}$ forward passes). Therefore, modern methods employ efficient but approximate **estimators** of the CMA score itself. In this

work, we consider Edge Attribution Patching (EAP; Syed et al., 2023) and its variants (Hanna et al., 2024) due to their ubiquity in the literature (Zhang et al., 2026; Mondorf et al., 2025; Nikankin et al., 2025) and their state-of-the-art performance in identifying sparse edge-level circuits (Syed et al., 2023; Hanna et al., 2024). These methods approximate the intervention $M_e(x) \leftarrow M_e(x_{\text{corr}})$ using gradient information. However, as these estimators are approximate and rely on per-input information to approximate the population-level effect of an intervention, they may introduce approximation noise that compounds the intrinsic variance of the CMA scores. We investigate four specific estimators of $S(e, x, x_{\text{corr}})$:

- **EAP:** A first-order Taylor approximation of $S$ that multiplies the gradient of the metric $\nabla \mathcal{L}(x)$ by the activation difference $M_e(x) - M_e(x_{\text{corr}})$ after intervention.

- **EAP-IG (inputs):** Uses integrated gradients, averaging $\nabla \mathcal{L}(x)$ over $m$ interpolation steps between $x$ and $x_{\text{corr}}$.

- **EAP-IG (activations):** Similar to the above, but integrates gradients w.r.t. intermediate activations, interpolating directly between clean and corrupted activation states.

- **Clean-corrupted:** Averages the gradient at two points only ($x$ and $x_{\text{corr}}$), without interpolation.

### 3.4. Measuring Stability

We decompose the instability of discovered circuits into two distinct sources: **(i) Variance (sampling sensitivity)** arises from relying on a finite dataset $D$ to approximate the population expectation. It measures the fluctuation of $\hat{S}$ when $D$ is resampled. High variance implies that the underlying distribution of per-input CMA scores is broad and the aggregate estimate unreliable. **(ii) Robustness (methodological sensitivity)** captures the sensitivity of the result to this specification of the counterfactual $x_{\text{corr}}$ (intervention strategy) and the hyperparameters $\Lambda$. To quantify these, we produce sets of $N$ circuits $\{C_1, \ldots, C_N\}$ under controlled variations and measure their structural and functional stability.

**Stability Metrics.** We report the following metrics across the generated circuit sets.

**(i) Structural stability (Jaccard index):** We quantify the structural spread of circuit estimates via the overlap between discovered edge sets $E_i, E_j$ corresponding to discovered circuits $C_i, C_j$. We report the mean and variance of the pairwise Jaccard index:

$$J(E_i, E_j) = \frac{|E_i \cap E_j|}{|E_i \cup E_j|}.$$

**(ii) Faithfulness:** We assess how well the different circuits

recover the model's behavior using the circuit error:

$$\text{CE}(C_i, f_\theta) = \frac{1}{|D|} \sum_{x \in D} \mathbb{1}[f_{C_i}(x) \neq f_\theta(x)]$$

and the KL divergence $D_{\text{KL}}(P_{f_\theta} || P_{f_{C_i}})$ averaged over $D$. We report mean $\mu$, variance $\sigma^2$, coefficient of variation $CV$ of both CE and $D_{\text{KL}}$. In all experiments, KL divergence and circuit error are highly correlated; we report the latter in the main part and the former in the appendices.

## 4. Experimental Setup

**Tasks and Datasets.** We follow the setup in Hanna et al. (2024) and use three standard interpretability tasks consisting of clean/corrupted input pairs:

**(i) Indirect Object Identification (IOI)** (Wang et al., 2023), involving identifying indirect objects in narratives. We use the generator from Wang et al. (2023).

**(ii) Subject-Verb Agreement (SVA)** (Newman et al., 2021), involving predicting the verb form that agrees with a singular or plural noun. We adapt the generator from Warstadt et al. (2020) to create pairs of singular/plural nouns only. Prompt paraphrasing was not implemented for this task due to the simplicity of the prompt.

**(iii) Greater-Than** (Hanna et al., 2023), involving predicting a year numerically greater than the one provided in the prompt. We use the dataset and the generator from Hanna et al. (2023) for distribution shifts.

We use the standard evaluation metrics of logit difference for IOI and SVA, and probability difference for Greater-Than.

**Models.** We conduct experiments across three language models: **gpt2-small** (Radford et al., 2019), selected as a foundational MI benchmark used in the original EAP, EAP-IG and ACDC studies; **Llama-3.2-1B** (AI@Meta, 2024), to test generality on a larger, modern architecture; and its instruction-tuned variant, **Llama-3.2-1B-Instruct**, as fine-tuning may impact the stability of causal mechanisms (Jain et al., 2024; Prakash et al., 2024).

**Perturbation Strategies.** We isolate sources of instability through specific regimes:

- **Data resampling:** We estimate sampling variance via bootstrap. We generate $n = 100$ datasets by resampling with replacement from $D$ and re-running the full discovery pipeline.

- **Distribution shifts:** We assess generalization using new datasets drawn from the same meta-distribution (meta-dataset) or by paraphrasing input prompts (Reprompting).

- **Intervention definition:** We investigate how the definition of the counterfactual $x_{\text{corr}}$ impacts discovery. Instead of a fixed corruption, we generate $x_{\text{corr}}$ by sampling different Gaussian noise to the token embedding. By varying the noise amplitude, we effectively alter the "strength" of the intervention, measuring how importance scores vary with the magnitude of the perturbation.

- **Methodological perturbations (robustness):** We test sensitivity to $\Lambda$ by varying the aggregation function (e.g., mean vs. median), the type of counterfactual (corrupted vs. mean patching), and comparing different base estimators (e.g., EAP vs. EAP-IG) on fixed data.

## 5. Results

We investigate empirically the stability of causal importance estimation and circuit discovery across sources of variability. Unless otherwise stated, we use the implementation from the EAP-IG library using its default hyperparameters.

### 5.1. Variability in Edge Scores

To distinguish between the natural variability of the model's mechanism and the error introduced by approximation methods, we first compute CMA exactly (computing Eq. 1) for each input sample and each edge in the computational graph. For this experiment, due to high computational costs, we restrict ourselves to the IOI dataset and gpt2-small. Figure 2 compares the mean and standard deviation (std) of these exact scores (blue) against the approximate EAP estimates (red). We observe two critical phenomena:

**Intrinsic Variability of CMA.** The causal effect of an edge is not stable across inputs. The blue distribution shows that edge scores exhibit a standard deviation often close to half their mean ($CV \approx 0.5$), confirming that edge importance display high variability and depends highly on the specific input-counterfactual pair.

**Approximation Instability.** EAP shifts the distribution and significantly increases the CV, with the standard deviation often exceeding the mean ($CV > 1$). As such, an edge's score is not consistent across samples. This indicates that gradient-based estimators introduce substantial approximation noise on top of the natural variance of the CMA estimand. Consequently, the signal-to-noise ratio for any given edge is low, making the identification of stable circuits from a finite sample statistically precarious.

### 5.2. Circuit Instability under Data Resampling

We next investigate how this instability (high variability of edge scores) propagates to the final circuit structure when varying the input dataset $D$. Figure 3 displays the functional performance (circuit error) and structural stability (Jaccard

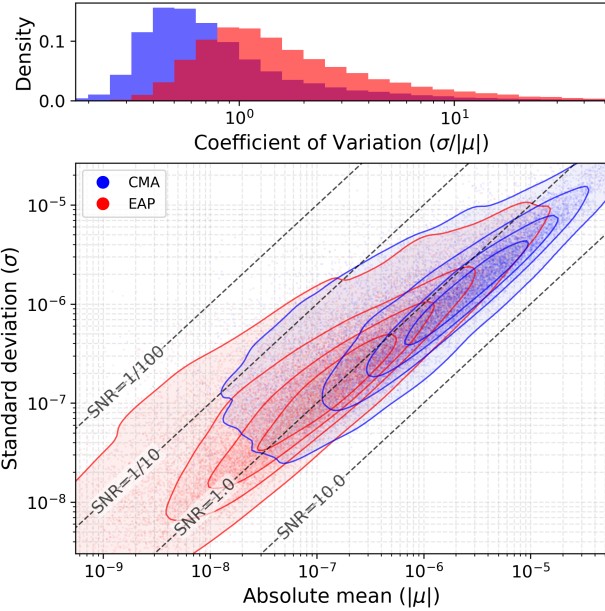

*Figure 2.* Distribution of edge scores for the Greater-Than task in gpt2-small across individual inputs. **Top:** The coefficient of variation ($CV = \sigma/|\mu|$) of edge scores across the dataset. A high CV indicates that the causal score of an edge displays marked instability between inputs. **Bottom:** Comparison of score distributions (mean vs. std) for exact edge ablation (blue) and EAP (red). While EAP generally has a lower mean and std than the underlying causal effect it attempts to approximate, the obtained edges have a consistently higher CV. Additionally, EAP introduces higher relative fluctuations in the mean and std across edges. A similar figure for the IOI task is available in Appendix 6.4.

index) of circuits discovered under different resampling strategies.

**Stability and Model Size.** We observe a notable degradation in stability for larger models. While gpt2-small yields relatively clustered results, Llama-3.2 (1B and Instruct) exhibits higher variability. This suggests that MI methods do not trivially scale; identifying reliable "circuits" in more capable models is significantly harder. Interestingly, instruction tuning (Llama-Instruct) does not significantly alter this stability profile compared to the base model.

**Multimodality**. For gpt2-small, the Jaccard index distribution is sometimes multimodal (visible in the split violins for bootstrap). This is consistent with non-identifiability, where multiple distinct circuits satisfy the scoring criteria, though other explanations (e.g., sensitivity to a few borderline edges) cannot be ruled out.

**Sensitivity to Sampling.** Table 1 quantifies the impact of the perturbation method. Bootstrap resampling, which mimics the effect of limited sample size, yields the lowest structural consistency (Jaccard $\mu = 0.561$) and highest variability ($CV = 0.335$). This confirms that the

*Table 1.* Aggregate statistics for circuit error and Jaccard index across resampling strategies (averaged over all models and tasks).

| Resampling Strategy | Circuit error | | Jaccard Index | |
|---|---|---|---|---|
| | $\mu$ | $CV$ | $\mu$ | $CV$ |
| Bootstrap | **0.440** | 0.123 | **0.561** | **0.335** |
| Meta-Dataset | 0.300 | 0.094 | 0.790 | 0.132 |
| Prompt Paraphrasing | 0.150 | **0.134** | 0.799 | 0.131 |

high variance of edge scores (Fig. 2) makes the aggregated mean $\hat{S}$ highly sensitive to the specific composition of the dataset. Conversely, shifting the meta-distribution (meta-dataset/paraphrasing) yields more stable results. This suggests that while the specific edges fluctuate with sampling noise (bootstrap), the general mechanism is somewhat more robust to semantic shifts in the prompt distribution.

The circuits discovered under bootstrap resampling also exhibit the highest average circuit error (0.440), indicating that the resulting circuits are not only structurally different but also less faithful to the original model's behavior, i.e., discovered circuits do not generalize well to small data variations. In contrast, using a meta-dataset or prompt paraphrasing results in more stable circuits, with higher Jaccard indices (resp. 0.790 and 0.799) and lower CVs.

### 5.3. Methodological Sensitivity: Hyperparameters

We next evaluate the robustness of circuit discovery to the value of hyperparameters. Figure 1 (in the introduction) provides a visual summary of how varying multiple parameters at once leads to a high diversity in circuits found in gpt2-small for the Greater-Than task.

Since the data signal is noisy, we hypothesize that the resulting circuit is heavily influenced by the choices of estimator $\mathcal{E}$ and aggregation $\mathcal{A}$. Table 2 confirms this for Llama-Instruct. In the Greater-Than task, changing the aggregation method of EAP-IG-inputs from "sum" to "median" and the patching method from "mean" to "patching" drops the Jaccard similarity to the median circuit to 0.086, effectively returning almost a disjoint subgraph. In IOI, the overlap between EAP-IG-inputs and Clean-corrupted is also negligible (0.071). This implies that different EAP variants are not converging on the same circuit, but are instead isolating different artifacts of the high-variance edge distribution.

### 5.4. Sensitivity to Counterfactual Choices

Finally, we explore how the definition of the intervention alters the results. As discussed in Section 3, CMA is defined relative to a specific counterfactual $x_{\text{corr}}$. In noisy intervention setups, $x_{\text{corr}}$ is generated by adding Gaussian noise to the token embedding. Varying the noise amplitude implies changing the experimental question: which components me-

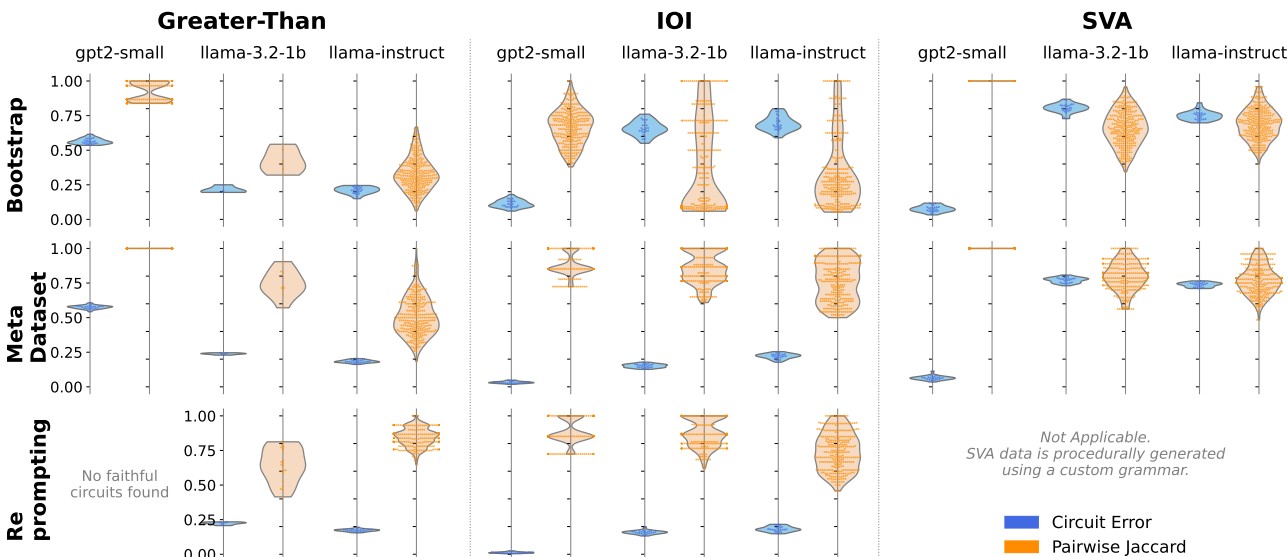

*Figure 3.* Stability of EAP-IG circuits across models and tasks. Each point represents one circuit discovered from a resampled dataset. **Blue:** Circuit error (lower is better). **Orange:** Pairwise Jaccard index (higher is better).

*Table 2.* Hyperparameter sensitivity in Llama-3.2-1B-Instruct. We report the circuit error (CErr), size, and Jaccard similarity to the median circuit (computed across all 7 rows) for varying EAP configurations. Results for other models are reported in the appendix.

| Parameters | Greater-Than | | | IOI | | | SVA | | |
|---|---|---|---|---|---|---|---|---|---|
| | CErr | Size | Jacc. to Median | CErr | Size | Jacc. to Median | CErr | Size | Jacc. to Median |
| EAP, sum, patching | 0.20 | 23 | 0.417 | 0.69 | 3 | 0.286 | 0.76 | 18 | 0.536 |
| EAP-IG-activations, sum, patching | 0.20 | 17 | 0.098 | 0.69 | 12 | 0.125 | 0.76 | 24 | 0.531 |
| EAP-IG-inputs, median, patching | 0.20 | 10 | 0.086 | 0.69 | 6 | 1.000 | 0.75 | 21 | 0.840 |
| EAP-IG-inputs, sum, mean | **0.19** | **28** | **1.000** | 0.72 | 7 | 0.182 | 0.73 | 24 | 0.960 |
| EAP-IG-inputs, sum, mean-positional | 0.41 | 33 | 0.298 | **0.82** | **6** | **1.000** | 0.73 | 22 | 0.808 |
| EAP-IG-inputs, sum, patching | 0.20 | 16 | 0.571 | 0.69 | 7 | 0.182 | **0.75** | **25** | **1.000** |
| Clean-corrupted, sum, patching | 0.20 | 16 | 0.419 | 0.69 | 9 | 0.071 | 0.76 | 16 | 0.577 |

diate the effect of small vs. large deviations in the input? While practitioners might hope the identified circuit is robust to reasonable variation in perturbation strength, our results show this is not the case. Figure 4 shows the trajec-

**Greater-Than**

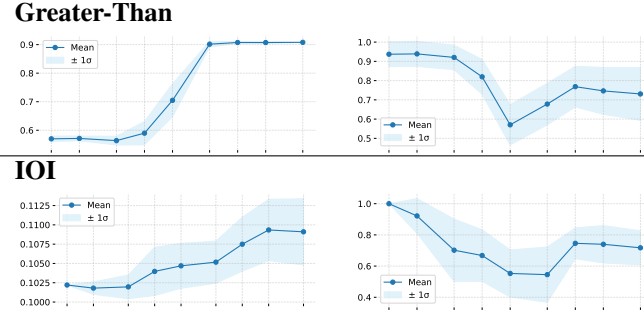

**IOI**

*Figure 4.* Effect of intervention definition (added noise amplitude) on circuit error (left) and pairwise Jaccard index (right) in gpt2-small. The amplitude parameter effectively redefines the counterfactual input $x_{\text{corr}}$, leading to changes in the identified mechanism. Noise amplitude varies in [0.01, 0.02, 0.05, 0.1, 0.2, 0.5, 1, 2, 5].

tories of circuit error and Jaccard index for gpt2-small as

the noise amplitude increases. We identify a critical regime (amplitude ≈ 0.2) where the CV for Jaccard index peaks. This demonstrates that the "circuit" is not invariant to the magnitude of the perturbation. As the intervention changes, the set of components identified as important shifts, further emphasizing that MI findings are relative to the precise definition of the counterfactual distribution.

## 6. Discussion

### 6.1. Summary of Findings

Our investigation traces the source of the observed instability through the causal analysis pipeline:

- **Intrinsic & Estimator Variance.** We distinguish two sources of instability: the fundamental estimand (causal effect of an edge) is not a constant but a random variable with high variance across inputs drawn from a distribution, and gradient-based estimators (EAP) amplify this variance, often yielding a signal-to-noise ratio below 1.

- **Aggregation Sensitivity.** As the underlying signal is noisy, the final circuit depends heavily on the specific sample used for aggregation. Circuits discovered from the same model on resampled data can exhibit low structural overlap, confirming that single-dataset results are statistically unreliable and not generalizable.

- **Dependence on Experimental Definition.** The discovered circuits are highly sensitive to the experimental design choices. We find that design choices in the estimation process and the definition of the counterfactual fundamentally create high structural variability in final circuits. This confirms that these methods do not identify a unique, population-level mechanism, but rather a structure conditioned on methodological choices.

It is worth distinguishing two related but distinct issues.

**Non-identifiability** is a theoretical property: the impossibility of uniquely recovering a circuit from observed data, even with infinite samples (Méloux et al., 2025).

**Estimator instability** (what we measure) is an empirical symptom: the observation that circuits change substantially under perturbation. High instability is consistent with non-identifiability, but does not prove it: some of the observed variability may stem from finite-sample noise or approximation error that could in principle be reduced. Our contribution is to quantify this instability and show that it is large enough to undermine current reporting practices, regardless of its root cause.

### 6.2. Recommendations for a Statistical MI

For future research to mitigate these risks, we propose the following recommendations:

**Report Stability.** We strongly advocate for the routine reporting of stability metrics alongside circuit discovery results. Specifically, we recommend that researchers report the variance of circuit structure and performance (e.g., the average pairwise Jaccard index and the CV of the circuit error) under bootstrap resampling of the input data. This practice, common in mature scientific fields (Efron & Tibshirani, 1986; Berengut, 2006), provides measures of uncertainty for the structural estimate. As a preliminary guideline, a mean pairwise Jaccard index above 0.8 under bootstrap resampling (with $n \geq 100$ resamples) could serve as a reasonable minimum bar for reporting a circuit as stable. If a circuit does not meet this threshold, practitioners should report caveats about their structural reliability. We stress that this is a tentative suggestion and that establishing principled thresholds requires broader community discussion.

**Quantify Estimator Uncertainty.** Given the sensitivity of circuit discovery to hyperparameters, it is crucial that researchers transparently report and justify their choices.

Ideally, a sensitivity analysis should be conducted to assess the impact of different hyperparameters on the discovered circuits. If a mechanism is only visible under a specific set of hyperparameters, this fragility must be disclosed.

**Characterize Intervention Sensitivity.** Instead of relying on a single fixed intervention (e.g., mean ablation), we recommend analyzing how the circuit changes as the counterfactual is varied. Sweeping intervention parameters (e.g., the noise amplitude) reveals whether a mechanism is invariant to the strength of the perturbation or specific to a certain regime. For example, reporting how circuit stability shifts around a noise level of 0.2 in gpt2-small can help distinguish between core mechanisms and localized artifacts.

### 6.3. Limitations

While our analysis identifies fundamental instabilities in circuit discovery, several limitations remain.

First, our circuit discovery analysis focuses on the EAP family and its variants. However, the fundamental sources of instability we identify are not specific to the EAP family. Any method that seeks to identify sparse circuits must (a) estimate per-input importance scores, which we show are intrinsically variable even under exact CMA, (b) aggregate these scores over a finite dataset, and (c) apply some discrete selection heuristic. Steps (b) and (c) are where small fluctuations get amplified into structural differences, regardless of how step (a) is implemented. Furthermore, the best performing non-EAP circuit discovery methods, such as NAP (Mueller et al., 2025), HAP (Gu et al., 2025), or RelP (Jafari et al., 2025), rely on CMA. A notable exception is GIM (Edin et al., 2025), which may exhibit different noise profiles and use selection rules that could act as implicit regularizers. However, these techniques face the same underlying challenge for steps (b) and (c). Characterizing their stability is an important direction for future work.

Second, while we established intrinsic variance via exact CMA, computational costs restricted this to gpt2-small on the IOI and Greater-Than tasks; generalizing this layer of instability to other models and tasks relies on approximation-based evidence.

Third, our study is limited to three classic MI tasks with relatively discrete linguistic rules; instability may manifest differently in fuzzier reasoning tasks or open-ended generations. Finally, our stability metrics treat all edges as equally important, whereas weighted stability metrics might reveal a stable "functional core" of the circuit despite a fluctuating periphery.

### 6.4. Future Directions

Our work opens up several avenues for future research. The high instability of discovered circuits suggests that instead

of seeking a single "true" circuit, it might be more fruitful to characterize a distribution over possible circuits.

**Probabilistic Circuit Discovery.** Since the underlying CMA scores are distributions, the output of an MI method could be a posterior distribution over graphs, rather than a single discrete subgraph. The set of bootstrapped circuits generated in this study serves as a first approximation of such a distribution. Future work could formalize this using Bayesian structure learning approaches.

**Decomposing Variance.** To improve methods' reliability, future work should aim to decompose the total observed variance into estimator variance (noise from the gradient estimation) and intrinsic variance (true fluctuations in the mechanism across inputs). Reducing estimator variance is an engineering challenge for better approximations, while high intrinsic variance suggests fundamental limits to the universality of specific mechanisms.

**Stability-Aware Optimization.** Our findings motivate the development of objectives that explicitly optimize for stability. Rather than selecting edges solely based on faithfulness (magnitude of effect), future algorithms could penalize the variance of the edge score across the dataset, prioritizing components that serve as reliable mediators across the dataset, bootstrap resamples or noise perturbations. The aggregation step could also be made more stable, such as replacing the empirical mean with shrinkage estimators or robust statistics.

While the statistical perspective we have adopted is broadly applicable to circuit discovery methods, we encourage the community to adopt similar stability analyses for other interpretability techniques to build a more complete picture of the reliability of MI findings. Despite recurrent analogies to other sciences like *neuroscience* (Barrett et al., 2019), *biology* (Lindsey et al., 2025), or *physics* (Allen-Zhu & Li, 2023; Allen-Zhu, 2024) of neural networks, the field of MI remains in its early stages. We believe that embracing a statistical estimation framing and its standards of rigor regarding uncertainty quantification is an important step toward becoming a more robust and rigorous field.

## Acknowledgements

This work was conducted within French research unit UMR 5217 and was partially supported by CNRS (grant ANR-22-CPJ2-0036-01) and by MIAI@Grenoble-Alpes (grant ANR-19-P3IA-0003). It was granted access to the HPC resources of IDRIS under the allocation 2025-AD011014834 made by GENCI.

## Impact Statement

This work aims to improve the scientific rigor and reliability of Mechanistic Interpretability (MI). As MI techniques are increasingly proposed for safety auditing, model alignment, and regulatory compliance, it is critical that these methods produce stable and statistically valid explanations. Our research highlights the risks of relying on unstable point-estimates, which can lead to unjustified confidence in a model's safety properties or internal mechanisms. By advocating for statistical robustness and best practices in circuit discovery, this work contributes to the development of more trustworthy AI systems and helps ensure that future interpretability tools provide a solid foundation for policy and safety decisions.

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

## Additional plots

**Varying multiple circuit-finding parameters at once**

The results in this section pertain to the experiment of Figure 1, in which multiple circuit-finding parameters are varied at once.

5 contains the pairwise Jaccard index for all 125 circuits.

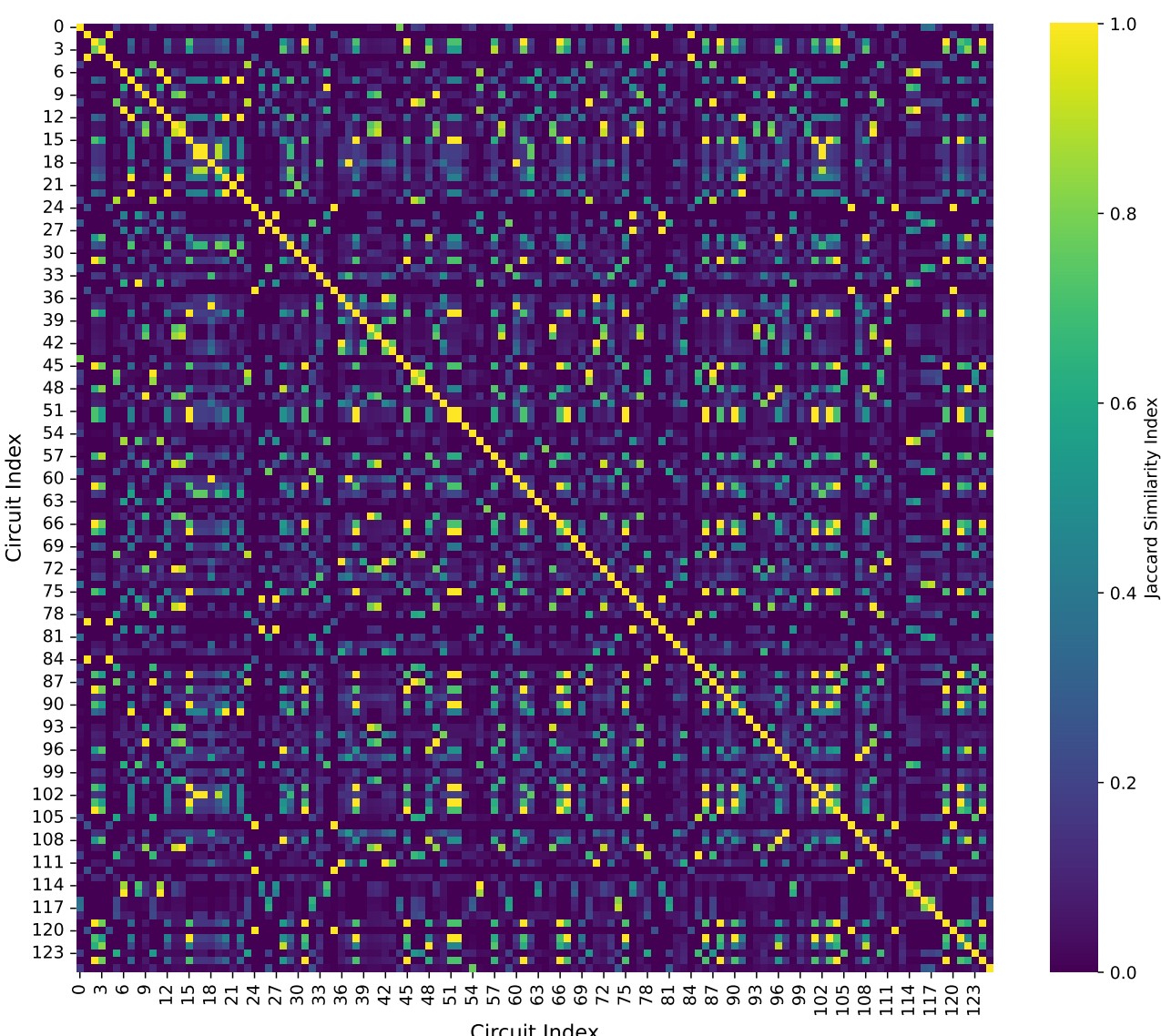

*Figure 5.* Full heatmap of the pairwise Jaccard index between circuits displayed in Figure 1 (circuits found in gpt2-small on the Greater-Than task while varying all parameters)

Figure 6 shows the distribution of edge selection frequencies across all circuits, and we report in Table 3 the top 50 most selected edges. Both results are computed on a larger number of circuits (330).

**Circuit stability summary**

Tables 4, 5, and 6 contain numerical values for the metrics reported in the violin plots of Figure 3.

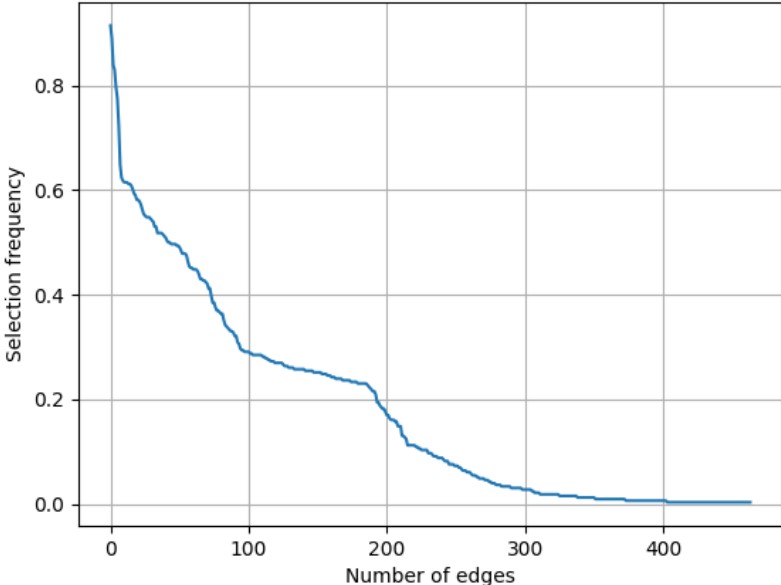

*Figure 6.* Edge selection frequency across circuits: Of the possible 32,491 edges in gpt2-small, only 464 are selected at least once in a circuit. However, most of these edges are seldom selected: approximately half of the edges (231) are present in at least 10% of all circuits, 94 edges are selected in at least 30% of all circuits, and only a few edges (7) are present in over 80% of all circuits.

| Rank | Edge | Frequency | Rank | Edge | Frequency |
|------|------|-----------|------|------|-----------|
| 0 | m10 → logits | 0.915 | 25 | a6.h4 → m6 | 0.552 |
| 1 | m9 → m10 | 0.891 | 26 | m2 → m3 | 0.548 |
| 2 | a9.h9 → logits | 0.839 | 27 | m1 → m8 | 0.548 |
| 3 | m3 → m4 | 0.830 | 28 | input → m0 | 0.548 |
| 4 | a8.h3 → a9.h9q | 0.797 | 29 | m8 → m11 | 0.545 |
| 5 | m5 → m10 | 0.778 | 30 | m2 → m6 | 0.542 |
| 6 | m3 → m5 | 0.724 | 31 | m6 → m7 | 0.539 |
| 7 | m8 → logits | 0.648 | 32 | m4 → m6 | 0.530 |
| 8 | m11 → logits | 0.624 | 33 | m7 → m10 | 0.530 |
| 9 | a8.h1 → logits | 0.618 | 34 | m3 → m7 | 0.518 |
| 10 | m1 → m2 | 0.615 | 35 | m3 → a8.h3k | 0.518 |
| 11 | m0 → m1 | 0.615 | 36 | a8.h8 → m10 | 0.518 |
| 12 | m0 → m3 | 0.615 | 37 | a9.h1 → logits | 0.518 |
| 13 | m7 → m8 | 0.612 | 38 | a8.h6 → m10 | 0.515 |
| 14 | m0 → m5 | 0.612 | 39 | m6 → m8 | 0.512 |
| 15 | a8.h10 → logits | 0.609 | 40 | a7.h7 → m9 | 0.509 |
| 16 | m1 → m3 | 0.603 | 41 | a7.h6 → logits | 0.503 |
| 17 | m10 → m11 | 0.594 | 42 | m5 → a7.h6k | 0.5 |
| 18 | a7.h6 → a8.h10q | 0.591 | 43 | a8.h6 → m11 | 0.5 |
| 19 | a7.h7 → logits | 0.581 | 44 | m3 → a7.h6k | 0.497 |
| 20 | m5 → m8 | 0.581 | 45 | a8.h6 → m8 | 0.497 |
| 21 | a8.h10 → a9.h9q | 0.579 | 46 | m8 → a9.h1q | 0.497 |
| 22 | a10.h2 → logits | 0.573 | 47 | m4 → m10 | 0.497 |
| 23 | m4 → m5 | 0.564 | 48 | m6 → a8.h3v | 0.494 |
| 24 | a9.h6 → m10 | 0.555 | 49 | a7.h6 → m11 | 0.494 |

*Table 3.* Label and frequency of the top 50 most selected edges. m denotes outputs of MLP layers; aX.Y denotes the output of attention head Y of layer X; q, k and v refer to the output of the respective attention matrices.

*Table 4.* Aggregated results from Figure 3 for bootstrap resampling.

| Model Name | Circuit Error | | | KL Divergence | | | Pairwise Jaccard Index | | |
|---|---|---|---|---|---|---|---|---|---|
| | $\mu$ | $\sigma^2$ | $CV$ | $\mu$ | $\sigma^2$ | $CV$ | $\mu$ | $\sigma^2$ | $CV$ |
| **Greater-Than** | | | | | | | | | |
| Llama-3.2-1B | 0.21 | $4.67 \cdot 10^{-4}$ | 0.10 | $6.91 \cdot 10^{-7}$ | $1.29 \cdot 10^{-14}$ | 0.16 | 0.42 | $5.93 \cdot 10^{-3}$ | 0.18 |
| Llama-3.2-1B-Instruct | 0.21 | $5.94 \cdot 10^{-4}$ | 0.12 | $6.43 \cdot 10^{-7}$ | $6.50 \cdot 10^{-16}$ | 0.04 | 0.33 | $1.36 \cdot 10^{-2}$ | 0.36 |
| **IOI** | | | | | | | | | |
| Llama-3.2-1B | 0.66 | $2.51 \cdot 10^{-3}$ | 0.08 | $5.48 \cdot 10^{-6}$ | $1.29 \cdot 10^{-13}$ | 0.07 | 0.39 | $1.07 \cdot 10^{-1}$ | 0.85 |
| Llama-3.2-1B-Instruct | 0.69 | $2.62 \cdot 10^{-3}$ | 0.07 | $9.26 \cdot 10^{-6}$ | $4.44 \cdot 10^{-13}$ | 0.07 | 0.34 | $6.72 \cdot 10^{-2}$ | 0.76 |
| gpt2-small | 0.11 | $7.32 \cdot 10^{-4}$ | 0.24 | $1.23 \cdot 10^{-6}$ | $8.80 \cdot 10^{-14}$ | 0.24 | 0.67 | $1.57 \cdot 10^{-2}$ | 0.19 |
| **SVA** | | | | | | | | | |
| Llama-3.2-1B | 0.80 | $1.02 \cdot 10^{-3}$ | 0.04 | $1.61 \cdot 10^{-5}$ | $4.02 \cdot 10^{-13}$ | 0.04 | 0.66 | $1.55 \cdot 10^{-2}$ | 0.19 |
| Llama-3.2-1B-Instruct | 0.75 | $1.04 \cdot 10^{-3}$ | 0.04 | $1.87 \cdot 10^{-5}$ | $3.97 \cdot 10^{-13}$ | 0.03 | 0.69 | $1.20 \cdot 10^{-2}$ | 0.16 |
| gpt2-small | 0.08 | $5.00 \cdot 10^{-4}$ | 0.29 | 0 | 0 | | 1.00 | 0 | 0.00 |

*Table 5.* Aggregated results from Figure 3 for meta-dataset resampling.

| Model Name | Circuit Error | | | KL Divergence | | | Pairwise Jaccard Index | | |
|---|---|---|---|---|---|---|---|---|---|
| | $\mu$ | $\sigma^2$ | $CV$ | $\mu$ | $\sigma^2$ | $CV$ | $\mu$ | $\sigma^2$ | $CV$ |
| **Greater-Than** | | | | | | | | | |
| Llama-3.2-1B | 0.24 | $3.06 \cdot 10^{-5}$ | 0.02 | $5.58 \cdot 10^{-7}$ | $3.56 \cdot 10^{-16}$ | 0.03 | 0.74 | $8.17 \cdot 10^{-3}$ | 0.12 |
| Llama-3.2-1B-Instruct | 0.18 | $1.05 \cdot 10^{-4}$ | 0.06 | $6.46 \cdot 10^{-7}$ | $1.31 \cdot 10^{-16}$ | 0.02 | 0.51 | $1.83 \cdot 10^{-2}$ | 0.27 |
| **IOI** | | | | | | | | | |
| Llama-3.2-1B | 0.15 | $1.67 \cdot 10^{-4}$ | 0.09 | $5.75 \cdot 10^{-7}$ | $6.68 \cdot 10^{-16}$ | 0.04 | 0.86 | $1.25 \cdot 10^{-2}$ | 0.13 |
| Llama-3.2-1B-Instruct | 0.22 | $3.30 \cdot 10^{-4}$ | 0.08 | $6.19 \cdot 10^{-7}$ | $1.53 \cdot 10^{-15}$ | 0.06 | 0.76 | $2.13 \cdot 10^{-2}$ | 0.19 |
| gpt2-small | 0.03 | $5.23 \cdot 10^{-5}$ | 0.22 | $4.72 \cdot 10^{-5}$ | $1.91 \cdot 10^{-12}$ | 0.03 | 0.88 | $5.75 \cdot 10^{-3}$ | 0.09 |
| **SVA** | | | | | | | | | |
| Llama-3.2-1B | 0.77 | $3.60 \cdot 10^{-4}$ | 0.02 | $1.54 \cdot 10^{-5}$ | $8.18 \cdot 10^{-14}$ | 0.02 | 0.80 | $1.06 \cdot 10^{-2}$ | 0.13 |
| Llama-3.2-1B-Instruct | 0.74 | $2.52 \cdot 10^{-4}$ | 0.02 | $1.84 \cdot 10^{-5}$ | $2.05 \cdot 10^{-13}$ | 0.02 | 0.77 | $1.07 \cdot 10^{-2}$ | 0.13 |
| gpt2-small | 0.06 | $2.18 \cdot 10^{-4}$ | 0.23 | 0 | 0 | | 1.00 | 0 | 0.00 |

*Table 6.* Aggregated results from Figure 3 for prompt paraphrasing.

| Model Name | Circuit Error | | | KL Divergence | | | Pairwise Jaccard Index | | |
|---|---|---|---|---|---|---|---|---|---|
| | $\mu$ | $\sigma^2$ | $CV$ | $\mu$ | $\sigma^2$ | $CV$ | $\mu$ | $\sigma^2$ | $CV$ |
| **Greater-Than** | | | | | | | | | |
| Llama-3.2-1B | 0.22 | $7.77 \cdot 10^{-5}$ | 0.04 | $7.09 \cdot 10^{-7}$ | $2.05 \cdot 10^{-15}$ | 0.06 | 0.64 | $1.42 \cdot 10^{-2}$ | 0.19 |
| Llama-3.2-1B-Instruct | 0.17 | $7.46 \cdot 10^{-5}$ | 0.05 | $5.43 \cdot 10^{-7}$ | $1.04 \cdot 10^{-16}$ | 0.02 | 0.85 | $4.20 \cdot 10^{-3}$ | 0.08 |
| **IOI** | | | | | | | | | |
| Llama-3.2-1B | 0.16 | $1.66 \cdot 10^{-4}$ | 0.08 | $5.42 \cdot 10^{-7}$ | $9.45 \cdot 10^{-16}$ | 0.06 | 0.88 | $1.01 \cdot 10^{-2}$ | 0.11 |
| Llama-3.2-1B-Instruct | 0.18 | $3.44 \cdot 10^{-4}$ | 0.10 | $6.06 \cdot 10^{-7}$ | $1.43 \cdot 10^{-15}$ | 0.06 | 0.74 | $1.80 \cdot 10^{-2}$ | 0.18 |
| gpt2-small | 0.01 | $2.27 \cdot 10^{-5}$ | 0.40 | $4.31 \cdot 10^{-5}$ | $1.42 \cdot 10^{-12}$ | 0.03 | 0.89 | $7.66 \cdot 10^{-3}$ | 0.10 |

**Variability of exact CMA on the Greater-Than task**

Figure 7 contains additional results for the experiment described in 5.1: the experiment is now performed on the IOI task rather than Greater-Than.

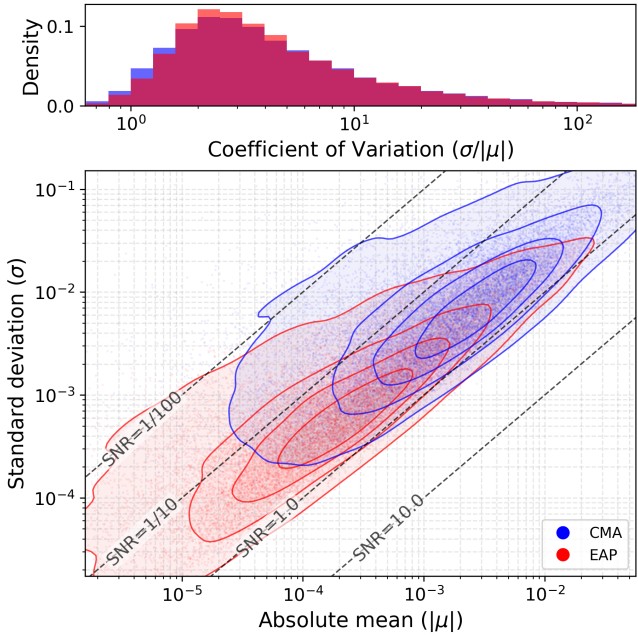

*Figure 7.* Distribution of edge scores for the IOI task in gpt2-small across individual inputs. **Top:** The coefficient of variation ($CV = \sigma/|\mu|$) of edge scores across the dataset. A high CV indicates that the causal score of an edge displays marked instability between inputs. **Bottom:** Comparison of score distributions (mean vs. std) for exact edge ablation (blue) and EAP (red). While EAP generally has a lower mean and std than the underlying causal effect it attempts to approximate, the obtained edges have a consistently higher CV. Additionally, EAP introduces higher relative fluctuations in the mean and std across edges..

**Hyperparameter sensitivity results on all models**

Table 7 is a more detailed version of Table 2, which also reports KL divergence. Tables 8 and 9 contain the equivalent data for Llama-3.2-1B (non-instruct) and gpt2-small, respectively.

*Table 7.* Detailed results for Table 2, including KL divergence.

| Parameters | Greater-Than | | | | IOI | | | | SVA | | | |
|---|---|---|---|---|---|---|---|---|---|---|---|---|
| | CErr | KL-Div | Size | Jacc. to Median | CErr | KL-Div | Size | Jacc. to Median | CErr | KL-Div | Size | Jacc. to Median |
| EAP, sum, patching | 0.20 | $6.4 \cdot 10^{-7}$ | 23 | 0.417 | 0.69 | $9.1 \cdot 10^{-6}$ | 3 | 0.286 | 0.76 | $1.9 \cdot 10^{-5}$ | 18 | 0.536 |
| EAP-IG-activations, sum, patching | 0.20 | $6.4 \cdot 10^{-7}$ | 17 | 0.098 | 0.69 | $9.1 \cdot 10^{-6}$ | 12 | 0.125 | 0.76 | $1.9 \cdot 10^{-5}$ | 24 | 0.531 |
| EAP-IG-inputs, median, patching | 0.20 | $6.4 \cdot 10^{-7}$ | 10 | 0.086 | 0.69 | $9.1 \cdot 10^{-6}$ | 6 | 1.000 | 0.75 | $1.9 \cdot 10^{-5}$ | 21 | 0.840 |
| EAP-IG-inputs, sum, mean | **0.19** | $7.1 \cdot 10^{-7}$ | **28** | **1.000** | 0.72 | $9.3 \cdot 10^{-6}$ | 7 | 0.182 | 0.73 | $1.6 \cdot 10^{-5}$ | 24 | 0.960 |
| EAP-IG-inputs, sum, mean-positional | 0.41 | $5.7 \cdot 10^{-6}$ | 33 | 0.298 | **0.82** | $1.7 \cdot 10^{-5}$ | **6** | **1.000** | 0.73 | $1.7 \cdot 10^{-5}$ | 22 | 0.808 |
| EAP-IG-inputs, sum, patching | 0.20 | $6.4 \cdot 10^{-7}$ | 16 | 0.571 | 0.69 | $9.1 \cdot 10^{-6}$ | 7 | 0.182 | **0.75** | $1.8 \cdot 10^{-5}$ | **25** | **1.000** |
| clean-corrupted, sum, patching | 0.20 | $6.4 \cdot 10^{-7}$ | 16 | 0.419 | 0.69 | $9.1 \cdot 10^{-6}$ | 9 | 0.071 | 0.76 | $1.9 \cdot 10^{-5}$ | 16 | 0.577 |

*Table 8.* Comparison of the circuits found in Llama-3.2-1B, using a similar setup to that of Table 2.

| Parameters | Greater-Than | | | | IOI | | | | SVA | | | |
|---|---|---|---|---|---|---|---|---|---|---|---|---|
| | CErr | KL-Div | Size | Jacc. to Median | CErr | KL-Div | Size | Jacc. to Median | CErr | KL-Div | Size | Jacc. to Median |
| EAP, sum, patching | - | - | - | - | 0.64 | $5.4 \cdot 10^{-6}$ | 7 | 0.400 | 0.80 | $1.6 \cdot 10^{-5}$ | 16 | 0.355 |
| EAP-IG-activations, sum, patching | - | - | - | - | 0.64 | $5.4 \cdot 10^{-6}$ | 117 | 0.042 | 0.80 | $1.6 \cdot 10^{-5}$ | 28 | 0.421 |
| EAP-IG-inputs, median, patching | - | - | - | - | 0.65 | $5.4 \cdot 10^{-6}$ | 11 | 0.385 | 0.80 | $1.6 \cdot 10^{-5}$ | 24 | 0.923 |
| EAP-IG-inputs, sum, mean | - | - | - | - | 0.67 | $5.4 \cdot 10^{-6}$ | 5 | 0.714 | 0.75 | $1.4 \cdot 10^{-5}$ | 26 | 1.000 |
| EAP-IG-inputs, sum, mean-positional | - | - | - | - | 0.77 | $8.8 \cdot 10^{-6}$ | 8 | 0.500 | 0.69 | $1.5 \cdot 10^{-5}$ | 25 | 0.962 |
| EAP-IG-inputs, sum, patching | 0.23 | $6.0 \cdot 10^{-7}$ | 21 | - | **0.65** | $5.4 \cdot 10^{-6}$ | **7** | **1.000** | **0.80** | $1.6 \cdot 10^{-5}$ | **26** | **1.000** |
| clean-corrupted, sum, patching | - | - | - | - | 0.59 | $5.2 \cdot 10^{-6}$ | 448 | 0.016 | 0.80 | $1.6 \cdot 10^{-5}$ | 16 | 0.355 |

*Table 9.* Comparison of the circuits found in gpt2-small, using a similar setup to that of Table 2.

| Parameters | IOI | | | | SVA | | | |
|---|---|---|---|---|---|---|---|---|
| | CErr | KL-Div | Size | Jacc. to Median | CErr | KL-Div | Size | Jacc. to Median |
| EAP, sum, patching | 0.10 | $1.2 \cdot 10^{-6}$ | 12 | 0.391 | 0.06 | 0 | 1 | 1.000 |
| EAP-IG-activations, sum, patching | 0.10 | $1.3 \cdot 10^{-6}$ | 5 | 0.042 | 0.05 | 0 | 21 | 0.000 |
| EAP-IG-inputs, median, patching | 0.11 | $1.2 \cdot 10^{-6}$ | 20 | 1.000 | 0.06 | 0 | 1 | 1.000 |
| EAP-IG-inputs, sum, mean | 0.12 | $1.3 \cdot 10^{-6}$ | 20 | 1.000 | 0.07 | $3.2 \cdot 10^{-6}$ | 1 | 1.000 |
| EAP-IG-inputs, sum, mean-positional | 0.14 | $2.1 \cdot 10^{-5}$ | 21 | 0.783 | 0.08 | $1.6 \cdot 10^{-5}$ | 1 | 1.000 |
| EAP-IG-inputs, sum, patching | **0.11** | $1.2 \cdot 10^{-6}$ | **20** | **1.000** | **0.06** | 0 | **1** | **1.000** |
| EAP-IG-inputs, sum, zero | - | - | - | - | 0.00 | 0 | 1 | 1.000 |
| clean-corrupted, sum, patching | 0.11 | $1.2 \cdot 10^{-6}$ | 19 | 0.696 | 0.06 | 0 | 1 | 1.000 |

## Noise experiment CVs

Figure 8 reports the CV of the faithfulness metrics for the noise experiments described in Section 5.3 and Figure 4.

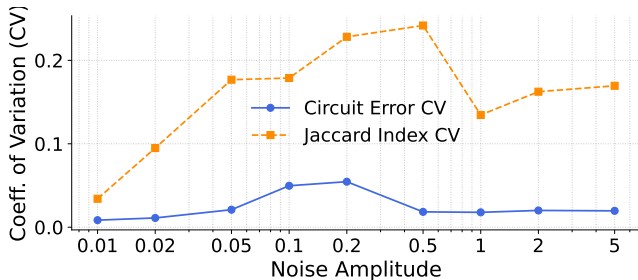

*Figure 8.* CV of circuit metrics for different noise amplitudes in gpt2-small, averaged across tasks.

## Detailed noise results

Table 10 is a more detailed equivalent of Table 4, reporting KL divergence in addition to other metrics.

*Table 10.* Detailed results for Table 4, including KL divergence. Values are plotted for noise amplitudes in [0.01, 0.02, 0.05, 0.1, 0.2, 0.5, 1, 2, 5].

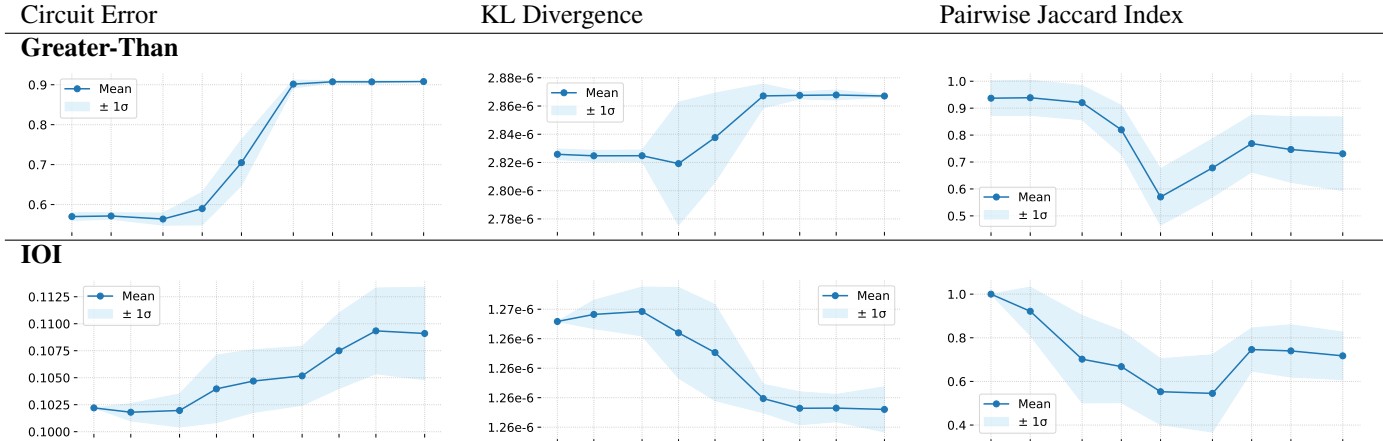

