# OpenReview forum: "Mechanistic Interpretability as Statistical Estimation: A Variance Analysis"
_ICML.cc/2026/Conference — ICML 2026 regular_

### Official Review · Reviewer_njky · 2026-03-08

**Soundness:** 3
**Presentation:** 4
**Significance:** 4
**Originality:** 4
**Overall Recommendation:** 5
**Confidence:** 4

**Summary:**

The paper argues that mechanistic interpretability, especially circuit discovery, should be treated as a statistical estimation problem rather than a deterministic discovery procedure. It traces instability back to the underlying causal mediation analysis scores, shows that approximation methods such as EAP and EAP-IG can amplify this instability, and then demonstrates that circuit extraction is highly sensitive to resampling, perturbation choice, and hyperparameters. The main contribution is therefore a variance-centered analysis of MI pipelines, along with recommendations to report stability metrics and adopt more statistically rigorous practices.

**Compliance With Llm Reviewing Policy:**

Affirmed.

**Final Justification:**

My final recommendation is accept, based on the paper’s strong empirical contribution, its relevance to the mechanistic interpretability (MI) community, and the authors’ thorough rebuttal, which addressed my main concerns and clarified the scope of the claims.

The paper tackles a timely and important question: the reliability and stability of circuit discovery in mechanistic interpretability. Rather than proposing a new method, it provides a systematic investigation of instability across the full pipeline, from causal mediation analysis (CMA) scores to final circuit extraction. This diagnostic perspective is valuable, as it shifts attention toward the scientific validity of interpretability results rather than solely improving performance. The empirical study is extensive and carefully designed, covering multiple tasks, models, and sources of variability. These aspects support strong evaluations in soundness and significance.

Regarding originality, while the core conceptual framing—viewing circuit discovery as a statistical estimation problem—builds on standard statistical principles, the paper’s contribution lies in making this perspective concrete through a large-scale empirical study. Importantly, the results go beyond general expectations by quantifying the magnitude and structure of instability in current MI pipelines. For example, the finding that instability persists through aggregation and leads to substantial structural variation in final circuits, as well as the identification of a very small stable core relative to the full circuit, provide actionable insights that are not obvious without systematic measurement.

My initial concerns focused on (1) the generality of the conclusions beyond the EAP/EAP-IG family, (2) the limited scope of exact CMA analysis, and (3) the distinction between estimator instability and deeper issues such as non-identifiability. The rebuttal addressed these points convincingly. In particular, the extension of exact CMA analysis to additional tasks strengthens the claim that high variance is intrinsic to the underlying estimand, while the clarification of how aggregation and selection amplify instability improves the generality of the conclusions. The added analysis of stable cores and the discussion of practical thresholds and variance-reduction strategies further enhance the paper’s usefulness.

In response to the Area Chair’s question regarding whether the findings are “expected,” my view is that while the existence of variability follows from basic statistical reasoning, the extent, structure, and consequences of this variability in MI pipelines are not obvious a priori. The paper provides concrete quantitative evidence showing that instability is both large and structurally impactful, which materially changes how one should interpret reported circuits. This elevates the contribution beyond a reminder of known principles and justifies its inclusion as a full paper rather than a position piece.

Some limitations remain. The empirical study is still centered on specific tasks and models, and exact CMA remains computationally constrained to smaller models. As a result, claims about universality should be interpreted with appropriate caution. Additionally, the paper is primarily diagnostic and offers limited concrete solutions beyond best practices and initial suggestions for variance reduction.

Overall, the rebuttal fully addressed my main concerns and strengthened my confidence in the contribution. I believe the paper provides an important and timely contribution that will encourage more rigorous and reliable practices in mechanistic interpretability research, and I support its acceptance.

**Key Questions For Authors:**

How much of the main conclusion depends specifically on the EAP/EAP-IG family, versus being expected to hold for other circuit discovery approaches as well? A broader empirical check would strengthen the paper.

Since exact CMA is only analyzed on gpt2-small and IOI, how confident should readers be that the same intrinsic variance phenomenon holds comparably in larger models and other tasks?

Did you examine whether unstable circuits still share a smaller stable “core” of important edges, even when full-graph Jaccard similarity is low?

Can the authors provide more guidance on what level of stability should be considered acceptable in practice for MI claims?

Do the authors see any promising variance-reduction strategies beyond reporting stability, such as regularized aggregation or stability-aware circuit selection?

**Limitations:**

yes

**Strengths And Weaknesses:**

Strengths
Timely and important question: it examines the scientific reliability of mechanistic interpretability rather than proposing yet another circuit-finding method.
Clear conceptual contribution: reframing circuit discovery as a statistical estimation problem is insightful and potentially valuable for the field.
Strong empirical angle: the paper studies multiple instability sources, including bootstrap resampling, dataset shifts, paraphrasing, perturbation design, and hyperparameter sensitivity.
Good practical takeaway: it gives concrete recommendations such as bootstrap reporting and stability metrics.
Broad relevance to MI: even though experiments focus on EAP-family methods, the central message likely matters more broadly for interpretability research.

Weaknesses
The empirical study is mostly centered on EAP / EAP-IG-style pipelines, so the generality of the conclusions to the full MI landscape is argued more than demonstrated.
Exact CMA analysis, which is central to the main claim about intrinsic variance, is only feasible on gpt2-small and one task, limiting how strongly that foundational claim can be generalized.
The paper is stronger as a diagnostic/meta-analysis than as a methodological advance; it identifies instability well, but offers limited concrete solutions beyond best practices.
Some conclusions may feel a bit stronger than the direct evidence supports, especially when extrapolating from the chosen tasks and models to MI as a whole.
Stability is mostly measured structurally (e.g., Jaccard overlap) and functionally via circuit error/KL, but there is less discussion of whether some unstable circuits may still share a stable functional core.

---

> ### Author Rebuttal · Authors · 2026-03-30
>
> We thank the reviewer for the strong endorsement of the question and the empirical approach, and for raising questions on stable cores, thresholds, and variance reduction that pushed us to strengthen the paper.
>
> Q1 (generality beyond EAP family): The EAP family and its derivatives consistently remain at the top of the MIB shared task benchmark (Mueller et al., 2025), and these are by far the most widely used methods in circuit discovery papers. Methods like warmstart, NAP, and EAP+IRL also explicitly rely on EAP. We now discuss this in Section 6.3, where we argue that the fundamental issue applies more broadly, for two main reasons:
> - Our exact CMA results show that the ground-truth causal importance of edges is itself volatile, which is a property of the model (and not of the method).
> - Any method that identifies sets of components must aggregate per-input scores and apply some thresholding, which we show amplifies instability. We specifically discuss NAP, HAP, RelP (which all rely on either EAP or CMA), and GIM (which does not use CMA, but faces the same aggregation/selection challenge). We agree and acknowledge that characterizing the specific stabilizing profile of non-CMA methods would be useful future work.
>
> Q2 (confidence in exact CMA, generalizing): In a new appendix, we have extended the CMA experiment of Figure 2 to the two other tasks. We find that our results on exact CMA not only still hold, but are even exacerbated in these other two tasks. Specifically:
> - On the previous experiment, 29% of CMA edge scores had a CV > 1 (66% for EAP). On the IOI task, this fraction increases to 97% (98% for EAP), and nearly 100% for the SVA task: almost all edges have inconsistent score signs across inputs.
> - The CMA CV distributions have a similar shape to the ones from the initial task, but shifted to the right (IOI: mode in [1, 2], SVA: mode in [2, 3]).
> - EAP shifts the numerical value of these exact CMA scores, increasing their mean and variance by ~x30 for IOI and ~x20 for SVA.
> - Interestingly, this shift affects values but not CV: the CV distribution is nearly identical for EAP CMA scores on both tasks, suggesting that the CMA-level instability is already so high that EAP does not amplify it much. Extending to larger models remains computationally prohibitive (each exact CMA run requires 2 * n_edges * n_samples forward passes). However, we argue that our claims remain broadly applicable:
> Even if the exact CMA scores had lower variance on some configurations, the approximation layer (EAP and variants) still adds its own instability on top, which we demonstrate across all three models and tasks.
> The EAP-based experiments show consistent instability patterns across all models and tasks, which is what one would expect if the underlying CMA scores are themselves variable. We have added text to the discussion in order to make this argument clearer and to specify which claims rest on exact CMA evidence vs. approximation-based evidence.
>
> Q3 (stable core of edges): This is an interesting question. In a new appendix, we now include a more detailed analysis of the circuits obtained in the Figure 1 experiment:
> - Out of 32,491 possible edges in gpt2-small, 464 are selected at least once in a circuit.
> - Among these edges, the selection frequency (across circuits) rapidly decreases. The top 7 most selected edges, present in 72-91% of circuits, are:
> 1. MLP 10 -> output
> 2. MLP 9 -> MLP 10
> 3. Head 9.9 -> output
> 4. MLP 3 -> MLP 4
> 5. Head 8.3 -> query input of head 9.9
> 6. MLP 5 -> MLP 10
> 7. MLP 3 -> MLP 5
> - Beyond that, the selection frequency decreases linearly and falls below 30% after 94 edges
> - Approximately half (231) of all selected edges are selected in >10% of circuits
> - The path among the top edges (early MLPs feed into the mid-layer ones, attention head 8.3 composes into head 9.9, and late MLPs project to the output) is consistent with prior accounts of the circuit for this task and model.
> - However, the stable core is very small (7/32491 edges) and is only present in ~80% of circuits. The circuits largely disagree on all other edges (how is information routed through intermediate layers? Which components have supporting roles?) Reporting stability metrics is therefore necessary to distinguish which parts of a discovered circuit reflect a robust mechanism rather than estimation artifacts.
>
> Q4 (acceptable stability thresholds): We believe that such stability thresholds ultimately require community consensus. Tentatively, we now suggest in Section 6.2 that a mean pairwise Jaccard index >0.8 under bootstrap (n > 100) could be a preliminary bar. Found circuits that do not meet this threshold should be accompanied by disclaimers about their robustness.
>
> Q5 (variance-reduction strategies beyond reporting): We discuss stability-aware optimization in Section 6.4, such as penalizing edge-score variance during selection. We have now expanded this to also mention regularized aggregation strategies as a complementary solution.

---

> > ### Author Rebuttal · Reviewer_njky · 2026-04-01
> >
> > The rebuttal thoroughly addresses my questions and significantly strengthens the paper.
> >
> > In particular, the discussion of generality beyond the EAP family is clearer and better justified. The argument that instability arises both from intrinsic variance in CMA and from aggregation/selection procedures is convincing, and the broader applicability to other circuit discovery methods is now more carefully scoped.
> >
> > The extension of the exact CMA analysis to additional tasks is especially valuable, as it reinforces the central claim that high variance is a fundamental property of the estimand rather than an artifact of a specific setup. The new analysis of stable cores is also insightful: it shows that while a small subset of edges is relatively consistent, the majority of the circuit structure remains unstable, which strengthens the paper’s main message.
> >
> > Finally, I appreciate the added discussion on practical guidance, including stability thresholds and variance-reduction strategies, which improves the paper’s usefulness for the community.
> >
> > Overall, these additions resolve my main concerns. The paper presents a compelling and important perspective on mechanistic interpretability as a statistical estimation problem, and I maintain my accept recommendation.

---

### Official Review · Reviewer_yYgF · 2026-03-12

**Soundness:** 3
**Presentation:** 3
**Significance:** 3
**Originality:** 2
**Overall Recommendation:** 4
**Confidence:** 3

**Summary:**

The paper addresses an important question for mechanistic interpretability: whether the circuits that are discovered are sufficiently stable to support strong scientific conclusions.
This central question is clearly stated and well motivated.
The authors also make a helpful distinction between local CMA scores, aggregation at the data level, and the final circuit selection.
By separating these stages, the paper highlights several points where variability can arise.
In particular, it argues that variance may be introduced through sampling, approximation, hyperparameter choices, and the definition of the counterfactual intervention.
Framing the issue this way clarifies why circuit discovery results may appear more definitive in a final visualization than they actually are in practice.

The experimental setup is also reasonably broad.
The authors examine three standard MI tasks (IOI, SVA, and Greater-Than) across three language models, while considering multiple potential sources of instability.
These include bootstrap resampling, meta-dataset shifts, prompt paraphrasing, changes in hyperparameters, and different intervention choices.
In addition, the paper evaluates circuits using both structural and functional metrics, specifically the Jaccard index and circuit error, which aligns well with the goal of analyzing stability alongside faithfulness.

**Compliance With Llm Reviewing Policy:**

Affirmed.

**Final Justification:**

The rebuttal addressed my concerns, and I increased my score by 1.

**Key Questions For Authors:**

Most of the following questions are already mentioned in the weaknesses above; here, I just made them more explicit.

1. How strong do you think the instability of CMA itself could generalize beyond IOI on gpt2-small? Right now, the exact-CMA analysis seems limited to IOI on gpt2-small because of computational cost. Can you provide additional exact-CMA evidence or any stronger supporting evidence? Otherwise, it would be better to clarify that this claim is only meant as a partial proof rather than a general conclusion.

2. To what extent do you think your results support the circuit discovery in general, instead of for the EAP family and related pipelines only? Since most experiments focus on the EAP family and variants, it would be nicer to be more precise about how broadly the conclusions apply.

3. Could you explain more in detail how you distinguish between estimator instability and non-identifiability of the underlying circuit? In the paper, there are several places that seem to move from variability to non-identifiability of the mechanism itself, which is a bit confusing.

4. Could you clarify the notations in sections 3.1 and 3.2, for example, Eq. 1? As mentioned before, the component $e$ does not appear on the right-hand side, making the definition hard to follow. Since moving from local score to global quantity is a key point of this paper, explicit and accurate notations are crucial for readability.

5. What concrete guidance could you provide for future practice? Are there any things one can do to obtain more trustworthy circuits?

**Limitations:**

yes

**Strengths And Weaknesses:**

***Strengths***

The paper discusses an interesting question in mechanistic interpretability: whether the discovered circuits are stable enough to depend on.
The question itself is clear, well-formulated, and nicely discussed.

In addition, the paper clearly distinguishes between local CMA scores, data-level aggregation, and the final circuit selection.
It clearly points out that the standard deviation could appear from data input, estimator approximation, hyperparameter selection, and the definition of counterfactual intervention.
This is a useful way to help identify why the reliability of circuit discovery is far lower than a single estimation.

The experiment covers multiple aspects: three standard MI tasks, three language models, and several sources of instability, including bootstrap resampling, meta-dataset shifts, prompt paraphrasing, hyperparameter changes, and intervention choices.
The selection of using the Jaccard index and circuit error as evaluation criteria also meets the goal of the paper.
The stability is studied in addition to faithfulness.

***Weaknesses***

It feels like the strongest evidence for the paper’s main claim is somewhat limited. While the authors argue that instability is intrinsic to CMA itself, the exact-CMA analysis was only applied to gpt2-small on IOI for computational reasons. In later discussions, the comparison mainly looks at the EAP family, and the broader claim across tasks and models is supported mainly through approximation-based evidence. Hence, it is not clear whether the observed high variance is caused by the CMA analysis itself or the approximate estimator. The conclusion about variablity of circuit discovery is stronger than what the results could show.

The presentation is overall understandable and well structured. But there are a few places where notations are confusing, and equations are not accurate. For example, in Eq. 1, the component $e$ does not appear on the right-hand side at all, and in section 3.2, there is a $\theta$ coming out of nowhere. The mixed usage of italic and bold is also sometimes confusing. In the end, the paper also does not distinguish clearly between "instability of estimators" and "non-identifiability of a circuit". Overall, these issues are minor.

The paper sounds more diagnostic rather than methodological. It neither introduces a new circuit discovery method nor suggests a new estimator or formal identifiability result. It mainly uses a statistical perspective to quantify the evaluation instability in a slightly more formal way. It offers a useful perspective, but only provides vague suggestions rather than novel workflow or criteria to concretely improve the pipeline.

---

> ### Author Rebuttal · Authors · 2026-03-30
>
> We thank the reviewer for recognizing the clarity of our decomposition and the breadth of the experiments, and for pointing out specific notation and clarity issues.
>
> On the limited CMA experiment: In a new appendix, we have extended the CMA experiment to the missing tasks on gpt2-small. We find that our previous results still hold, and are even exacerbated in these other two tasks:
> - In the initial experiment, 29% of CMA edge scores had a CV > 1 (66% for EAP). On the IOI task, this fraction increases to 97% (98% for EAP), and ~100% for SVA: almost all edges have inconsistent score signs across inputs.
> - The CMA CV distributions have a similar shape to the ones from the initial task, but shifted to the right (IOI: mode in [1, 2], SVA: mode in [2, 3]).
> - EAP shifts the numerical value of these exact CMA scores, increasing their mean and variance by ~x30 for IOI and ~x20 for SVA.
> - Interestingly, this shift affects values but not CV: the CV distribution is nearly identical for EAP CMA scores on both tasks, suggesting that the CMA-level instability is already so high that EAP does not amplify it much.
> Extending to larger models remains computationally prohibitive (each exact CMA run requires 2 * n_edges * n_samples forward passes). However, we argue that our claims remain broadly applicable:
> - Even if the exact CMA scores had lower variance on some configurations, the approximation layer (EAP and variants) still adds its own instability on top, which we show across all three models and tasks.
> - The EAP-based experiments show consistent instability patterns across all models and tasks, which is what one would expect if the underlying CMA scores are themselves variable. We have reworked the discussion to make this clearer.
>
> On generality beyond the EAP family: We now explicitly discuss this in 6.3. The fundamental sources of instability (variability of per-input CMA scores, finite-sample aggregation, discrete thresholding) apply to any circuit identification method, not just EAP. In addition, EAP is still the most commonly used family in the literature, and many of the more modern, SOTA methods as identified by the MIB shared task benchmark (Mueller et al., 2025, warmstart, NAP, EAP+ILR) are based on the CMA or EAP pipeline rather than entirely new techniques. GIM works differently and may have different noise profiles, but these few non-EAP techniques still face the same aggregation and selection challenges. We however acknowledge that future work could focus on quantifying their instability.
>
> On notation issues: We thank the reviewer for spotting these, and have improved the notation in multiple ways:
> - In Eq. 1, the component e is now on the right-hand side of the equation, written as $M_e(x)$ and $M_e(x_\text{corr})$ (this was also updated in Section 3.3).
> - In Section 3.2, we now introduce and describe the selection function $A$ and the hyperparameters $\Lambda$ before they appear in the equation. $\theta$ is now also introduced in Section 3.1.
> - We have changed the model notation from $M_\theta$ and $M_{C_i}$ to $f_\theta$ and $f_{C_i}$ in the faithfulness metrics to avoid confusion with the other usage of $M$ (for mediator activations).
>
> On non-identifiability vs. estimator instability: We have added a paragraph in Section 6.1 that distinguishes them. Non-identifiability is a theoretical property (impossibility of recovery even with infinite data), while estimator instability (previously loosely called “variance”) is the empirical phenomenon we measure. Our instability findings are consistent with non-identifiability but do not prove it: some of the observed variability could come from finite-sample noise or approximation error that could be reduced. Our contribution is to quantify this instability and show that it is large enough to undermine current reporting practices, regardless of the root cause.
>
> On the fact that the contribution is a diagnostic rather than methodological: We agree that this is a diagnostic contribution, which two other reviewers (5z4f, njky) highlight as valuable. We argue that MI needs this type of meta-analysis before developing principled solutions (instability cannot be reduced without first identifying its sources). Currently, the field largely ignores the issue, and we believe that quantifying sources of instability is a first necessary step. In terms of concrete guidance, our core recommendation is to routinely report stability metrics (bootstrap Jaccard, CV of circuit error) alongside any circuit finding. We have added a new threshold recommendation (Jaccard > 0.8 under n > 100 bootstrap resamples). However, this is tentative and needs to arise from community consensus. We now also discuss regularized aggregation as a concrete variance reduction strategy in Section 6.4. The deeper issue is that “more trustworthy” circuits may not exist as unique objects due to non-identifiability, in which case the right approach is to quantify and report how robust any given finding is.

---

> > ### Author Rebuttal · Reviewer_yYgF · 2026-03-31
> >
> > Thank you for the rebuttal. It addressed my main concerns. Although some broader claims should still be interpreted with appropriate caution, I believe the paper is stronger after rebuttal. So I am increasing my score from 3 to 4.

---

### Official Review · Reviewer_5z4f · 2026-03-14

**Soundness:** 3
**Presentation:** 3
**Significance:** 3
**Originality:** 3
**Overall Recommendation:** 5
**Confidence:** 4

**Summary:**

The paper takes a critical look at the reproducibility/stability of mechanistic interpretability (specifically, circuit discovery), pointing out several potential sources of variability. These sources include (1) the variance of estimated importance scores for each component, (2) the dependence of the population-level importance scores on design choices such as the input distribution and interventional distributions, and (3) the hyperparameter dependence of downstream aggregation methods.

Empirically, they investigate the role of each of these potential sources of variability in the circuit discovery pipeline, across three language models and three standard interpretability tasks. They find that, even when using an unbiased estimator, the estimated importance scores exhibit high variance (approximated via bootstrapping), and that this variance increases when using biased estimation methods such as Edge Attribution Patching (EAP); indicating that source (1) introduces large variability. Similarly, they show that this variability propagates to the final circuit structure. Then, they show that sources (2) and (3) also lead to large variability in findings, i.e., the final circuits are quite sensitive to hyperparameters and the corruption distribution. Lastly, the paper ends with a call to action to the mechanistic interpretability community, both encouraging more rigorous empirical work and laying out future directions suggested by their findings.

**Compliance With Llm Reviewing Policy:**

Affirmed.

**Final Justification:**

The paper makes a solid contribution towards setting up more rigorous "variability/stability"-based evaluations of mechanistic interpretability methods.

I agree with Reviewer RBdC's statement: "my current impression is that the paper is primarily reminding the MI community of principles it has been neglecting, in a careful and systematic way, rather than delivering unexpected insight." Also in agreement with Reviewer RBdC, it seems that this message fits better with a position paper track.

Nonetheless, I will maintain my raised score of "5" for a few reasons:
1. The content of the paper straddles the border between a position paper and a main track paper, and I do not want to penalize work that does not cleanly conform to this distinction. In particular, the paper does include a quite extensive empirical study that is beyond the scope of the typical position paper, and this study plays a central role in reinforcing the message.
2. The overall message is an important one at the time of writing - as in Reviewer RBdC's, the MI community is *indeed* neglecting certain fundamental principles in a problematic way. Given the importance of this message, I think it's more important for the paper to be published promptly rather than going through another round where they submit to "the right track".
3. Beyond the more position-paper style message, the paper provides practical insights that are not obvious a priori, though I agree with Reviewers RBdC and yYgF that these insights could be made in a more precise fashion. In particular, attributing overall instability to the variance of exact causal mediation analysis (CMA), instead of attributing it to approximation error (via the use of EAP-like methods), is not obvious a priori. This finding relies on the empirical study and has direct impacts on future research (e.g. focusing on variance reduction better than better approximation).

Overall, the author's responses addressed my main issues surrounding terminology - in agreement with Reviewer RBdC, I agree that these fixes were not merely cosmetic issues, and the proposed revisions reflect that the authors took seriously the use of precise statistical language. Further, the authors' plan to re-emphasize the fact that the paper is a systematic empirical study - rather than a methodological contribution - will enhance the final version.

**Key Questions For Authors:**

1. Can you provide a detailed description of how you plan to make the statistical language more accurate (see Weakness #1)?
2. Can you provide more explanation for the claims/intuitions mentioned in Minor Weakness #1?

**Limitations:**

Yes

**Strengths And Weaknesses:**

## Strengths
1. **Good motivation:** As someone coming from the field of causal structure learning, I completely agree with the author's main point, which they argue strongly: circuit discovery can and should be viewed as a statistical estimation problem, and the methods should be evaluated at a similar level of rigor. The concrete recommendations in Section 6.2 are especially appreciated, and the experiments give a good exemplar of how such analyses should be conducted.
2. **Good organization:** The paper progresses logically, and clearly lays out all steps of the circuit discovery pipeline. These steps provide a nice parallel structure throughout the paper, e.g. when discussing different sources of variability in Section 3.4 and when presenting the experimental results in Section 5.
3. **Extensive experiments:** The experiments touch upon all of the sources of variability mentioned by the authors, and the findings are consistent across different tasks and models. The metrics and visualizations are appropriate and compelling, and the evaluation strategies are a good template for future work.
## Weaknesses
1. **Imprecise statistical language:** Given the way that the paper positions itself as taking a more serious statistical viewpoint of circuit discovery, I found their use of statistical terms to be fairly misguided. Throughout, all sources of output variability in the outputs are referred to as "variance" (also present in the title!), but most of them are not variance in the actual statistical sense. It would be better to use a term with less precise statistical connotations, e.g. ambiguity, variability, reproducibility, stability etc. Indeed, the philosophy of the paper - analyzing the *entire* pipeline, and how it depends on several design choices - is closest in spirit to Veridical Data Science [1], and the authors may find it useful to adopt their terms like "stability", rather than abusing statistical terminology. This weakness was fairly distracting as a reader, especially given the paper's goals. I would be willing to raise my score if the authors could address this concern, as the paper is otherwise quite compelling.
### Minor Weaknesses
1. **Some tenuous claims/intuition:** In lines 315-317 (left column), the authors say that the multimodality of the Jaccard index implies that "the discovery process...vacillates between distinct, incompatible circuits", but I don't think this is the only explanation for the results; it would be good for the authors to consider this point more closely. Also, some intuitions (e.g. line 318-319, right column, "one might expect mediation results to not be affected by the choice of perturbation") are strange to me: the value of the importance score in Equation (1) clearly depends on the choice of perturbation. These are minor points, but the authors may want to check these intuition-related statements with different audiences.
### References
[1] Yu and Barber (2024), "_Veridical data science: The practice of responsible data analysis and decision making_."

---

> ### Author Rebuttal · Authors · 2026-03-30
>
> We thank the reviewer for the positive assessment of our motivation, organization, and experiments, and for the actionable suggestions on terminology and precision.
>
> W1 (imprecise statistical language, “variance” vs. “stability”): We thank the reviewer for raising this important point, and we agree that the previous vocabulary conflated the statistical quantity with the general sensitivity of the pipeline. We have gone through the entire manuscript and now only use “variance” for the former ($\sigma^2$ or CV), and “variability” or “(in)stability” to refer to the latter. Some examples include: abstract ("high intrinsic variability," "sources of instability"), Section 2 heading ("Identifying the Sources of Instability"), Section 5.1 heading ("Variability in Edge Scores"), Section 5.2 subheading ("Stability and Model Size"), and many instances in the text. Similarly, we have replaced the imprecise terms “local” and “global” with the clearer “per-input” and “population-level” (see our response to RBdC/N1). We have also added a citation to Yu & Kumbier (2020), "Veridical Data Science," in the introduction, as it is quite relevant to this work.
>
> Minor W1 (multimodality and non-identifiability): We have rephrased the sentence: "This is consistent with non-identifiability, where multiple distinct circuits satisfy the scoring criteria, though other explanations (e.g., sensitivity to a few borderline edges) cannot be ruled out."
>
> Minor W2 (intuition about perturbation invariance): We agree that the phrasing was misleading. The sentence is now "practitioners might hope the identified circuit is robust to reasonable variation in perturbation strength, but our results show this is not the case."
>
> Q1 (detailed description of terminology changes): See W1 above. In short, “variance” is now only used when referring to $\sigma^2$ or CV. We use “variability” as a neutral descriptive term for values varying across conditions, and “instability”/”stability”, which carry a judgement, for objects that are expected to be consistent but are not in practice (e.g. circuit instability under resampling). “Sensitivity” refers to a dependence on a specific factor (e.g. sensitivity to perturbation strength). These concepts are linked as follows: variability in edge scores leads to circuit instability, which reflects sensitivity to dataset composition, quantified by the high variance of per-input CMA scores.
>
> Q2 (more explanation for claims/limitations): See minor W1 and minor W2 above.

---

> > ### Author Rebuttal · Reviewer_5z4f · 2026-04-02
> >
> > Thank you for your thoughtful rebuttal!
> >
> > I am very happy to hear that you agree with the importance of the vocabulary, and I appreciate your distinction between the "statistical quantity" and "general sensitivity". The examples you provide have reassured me that the authors full understand the distinction and will successfully edit this language throughout the paper. I am pleased that the authors understand the relevance of the "veridical data science" framework to their work. I especially like the distinction between "variability" as a neutral term, "instability" as a term connected more to values which are expected to be consistent, and "sensitivity" as reflecting dependence on a specific factor - I think these distinctions are very valuable for the intended audience.
> >
> > Since the response has adequately addressed my concerns, I will raise my score from 4 to 5 (at the end of the Author-Reviewer discussion period, so that I can write an accurate "Final Justification").

---

### Official Review · Reviewer_RBdC · 2026-03-20

**Soundness:** 3
**Presentation:** 2
**Significance:** 2
**Originality:** 3
**Overall Recommendation:** 4
**Confidence:** 4

**Summary:**

The paper argues that circuit discovery in mechanistic interpretability (MI) should be viewed as a statistical estimation problem built on causal mediation analysis (CMA). The authors investigate the variance of CMA-based importance scores at multiple stages of the MI pipeline: from single-input edge scores, through gradient-based approximations, to the final extracted circuits. They consider multiple sources of variability, including resampling, distribution shifts, prompt paraphrasing, and hyperparameter choices, across three standard MI tasks and three language models. The main finding is that high variance is present at every stage. CMA scores are volatile across inputs and the resulting circuits are structurally unstable under small perturbations. The authors propose a number of best practices for the MI community, in light of their observations.

**Compliance With Llm Reviewing Policy:**

Affirmed.

**Final Justification:**

I appreciate the author's thorough and clear response to my follow up questions. It addressed my main concern, and I am convinced that the systematic evaluations the authors offer, although not surprising, is a diagnosis beyond a position paper. Therefore, I recommend a weak accept for the reason stated below.

I still do not find the findings (e.g., the bootstrap Jaccard average) to be surprising. Nevertheless, the authors' response has persuaded me that a systematic and quantified demonstration of the consequences of this overlooked statistical practice in MI constitutes a meaningful contribution and a valuable reminder for the community that describes the scale of the issue beyond a position paper. As noted in my original review, I consider this a timely and important topic; had the findings been more surprising or novel, and/or had the submitted draft more precisely situated the paper's contribution, I would have been inclined toward an accept. Given the current scope and the proposed revision plan, however, I find a weak accept the appropriate assessment. After the clarifications from the authors, I am also more confident about this revised assessment.

**Key Questions For Authors:**

Q1. In Sections 3.2 and 3.3, can the authors define more precisely what is meant by “local” and “global”?

Q2. What is the paper’s intended primary contribution: a new framework, or a systematic empirical study of instability in existing CMA-based circuit-discovery pipelines?

**Limitations:**

The paper includes a dedicated limitations subsection, which I appreciated. The stated limitations are reasonable, including the focus on the EAP family, and the restriction of exact-CMA analysis to gpt2-small. These limitations are important, and they further support presenting the paper in a more focused way as an empirical study of instability in current MI practice.

**Strengths And Weaknesses:**

**Strengths:**

S1. The problem the paper aims to address is timely and important. Any work toward principled and rigorous evaluation of mechanistic interpretability methods addresses a real gap, and the paper correctly identifies that stability and uncertainty quantification are largely underexplored in the MI literature.

S2. The experimental evaluation is thorough. The paper studies multiple tasks, multiple models, and several sources of perturbation.

---

**Major Weaknesses:**

W1. The main weakness in my opinion is the gap between what the paper promises and what it delivers. The introduction frames the work as proposing a statistical estimation framework for MI, but what follows is a collection of empirical observations about variance. There is no new estimator, no particularly informative formal framework or theoretical analysis, and no concrete method to reduce the variance. In fact, when the manuscript gets to its novel contributions, the paper does not seem to be presenting a framework, but rather observing that MI already is statistical estimation, like any empirical quantity, and then it measures the variance of these estimations. This is a legitimate empirical study, but the framing misrepresents the contribution.

W2. The formalization in Section 3.2 is vague and, to my reading, trivially restates that sample estimates have variance. Most statements in this subsection are true of any empirical estimate and not specific to MI. The conclusion that a circuit depends on the estimation pipeline is expected rather than novel. This subsection presents standard basics of what statistical estimations of target population/distribution quantities are as facts in MI-specific notation. This is confusing and does not constitute a framework contribution.

W3. The overall novel contribution is somewhat difficult to identify. As written, it is not fully clear which parts are intended as conceptual contribution versus standard setup for the empirical study.

W4. Overall, the paper aims to investigate a broad context, but I think the presentation would benefit from being more explicit that its main value lies in a systematic variance study that offers a collection of observations, rather than in introducing a substantially new formal framework. This reframing of the paper needs a substantial rewrite though.


---

**Minor Weaknesses:**

N1. The terminology in Section 3.2 (and again in 3.3) is unclear and appears unconventional. What do the authors mean by "local" and "global"? If "local" refers to per-input CMA scores and "global" refers to scores aggregated over a dataset, this is simply the sample-vs-population distinction and should be stated as such, to follow the standard terminology. The current language is confusing.

N2. Section 3.4 reads as experimental protocol, yet it is placed within "Formal Setup". This could give the impression that these standard methodological choices are being claimed as framework contributions. These are well-known tools. This section likely belongs in Section 4.

N3. The discussion is unusually long relative to the rest of the paper, and the mitigation advice would be clearer as a more focused separate section.

---

> ### Author Rebuttal · Authors · 2026-03-30
>
> We thank the reviewer for acknowledging the timeliness of the problem and the thoroughness of the experiments, and for suggesting concrete improvements to the framing and presentation.
>
> W1 (gap between promised framework and delivered contribution): We respectfully disagree that the paper “promises a framework”. Given the title (“A variance analysis”), the abstract (which describes decomposing sources of instability and advocating for stability metrics), and the contribution paragraph in the introduction, we believe the paper already presents itself as a systematic empirical study rather than a methodological contribution. We do not claim to propose a new estimator or a novel formal framework. We could only identify two sentences whose phrasing could give a different impression and have revised both (“MI should be reframed as a problem of statistical inference” -> “These issues call for applying standard tools of statistical inference to MI.”, and “the statistical framework we have proposed” -> “the statistical perspective we have adopted”). The objects studied in MI (sparse subgraphs extracted from a multi-stage pipeline) are complex enough that showing where instability enters/how much there is/how it propagates through the pipeline is a strong contribution on its own. Our paper produces evidence that applying basic statistics to MI is necessary (the field is largely not doing it currently), and we also provide a template for how to do it.
>
> W2 (section 3.2 trivially restates that estimates have variance): The statistical content of 3.2 is indeed not new, but the application to the field and resulting insights are. We have added to the beginning of the section: "The following formulation relies on standard concepts from statistical estimation, which we make explicit here. These standard concepts are rarely applied in MI practice: circuit discovery results are seldom reported with uncertainty estimates or stability analyses."
>
> W3 (it is unclear which parts are conceptual or part of the setup): We have clarified this:
> - We renamed Section 3 (“Formal Setup”) to “Formal Background” (to signal that the tools and notation it introduces already exist)
> - We moved the perturbation strategies to Section 4 (experimental setup).
> The contributions are now explicitly framed as (1) a systematic empirical study across the MI pipeline, and (2) concrete recommendations for more rigorous practice.
>
> W4 (reframing needed): We have revised the two sentences that could be misread (see W1), but given that the rest of the paper already presents the work as an empirical study, we do not think a substantial rewrite is warranted.
>
> N1 (local/global terminology): Thank you for the comment. We now use the more precise “per-input” and “population-level” throughout the paper instead of “local”/”global”.
>
> N2 (section 3.4 placement): As mentioned above, we moved the perturbation strategies to Section 4. The metric definitions (Jaccard, circuit error, and KL-divergence) are still in Section 3.4, which is now titled “Measuring Stability”, as these define mathematical quantities and are part of the formal background. We also changed the section title as mentioned in W3 to reduce the concern that standard tools were presented as contributions.
>
> N3 (length of the discussion): We have tightened the discussion, but have kept the core intact, since multiple reviewers (5z4f, njky) found the recommendations valuable.
>
> Q1: See N1 above.
>
> Q2: The primary contribution of the work is a systematic empirical study, which is now unambiguous in the revised text.

---

> > ### Author Rebuttal · Reviewer_RBdC · 2026-04-04
> >
> > I thank the authors for the detailed responses.
> >
> > The proposed revisions address several of my original concerns. If executed as described, the revised manuscript would be substantially improved in its positioning and clarity. That said, I want to emphasize that fixing the terminology is not a minor cosmetic issue. A paper whose central thesis is that MI should adopt more rigorous statistical practices must itself use precise statistical language.
> >
> > ---
> >
> > My main remaining concern, however, is about the contribution. The authors have now clarified that the paper is a systematic empirical study, not a methodological or theoretical contribution. The experiments are thorough and the paper provides concrete evidence to point out the issue with how the MI community currently often neglects and skips the standard practices of any empirical study. The paper quantitatively documents the consequences of this neglect.
> >
> > These are all valuable. However, the central observations, that finite-sample estimates have variance, that approximations add noise, and that thresholding amplifies instability, are expected from basic statistical reasoning. The specific findings confirm what one would predict a priori, and the recommendations are well-established practices in empirical science. In other words, at a high level, the message is that in empirical MI research we are working with sample-based quantities, they depend on design choices, and they should be accompanied by stability/uncertainty analysis. Those are long-established standards in observational and experimental work. So my current impression is that the paper is primarily reminding the MI community of principles it has been neglecting, in a careful and systematic way, rather than delivering unexpected insight.
> >
> > This leaves me with a question I would appreciate the authors addressing directly: What main takeaway of this paper do you think would be unexpected for someone from the MI community who has a working knowledge of statistics? If no such unexpected finding exists, could you please clarify what distinguishes your contribution from a position paper (ICML has a separate position paper track)?
> >
> > ---
> >
> > Needless to say, I'm open to adjusting my score. I am genuinely happy with the quality of the responses so far. But the question above remains central to my assessment, and I would like to hear the authors' perspective before making a final decision.

---

> > > ### Author Response · Authors · 2026-04-05
> > >
> > > We thank the reviewer for this precise follow-up question and willingness to engage in further discussion.
> > >
> > > To answer the question directly, our contribution goes significantly beyond stating that statistical rigor matters.
> > >
> > > We first agree that the need for rigorous statistical practice is long-established. However, statistical inference remains largely absent from MI, in part because the objects being inferred (sparse subgraphs of a computational graph) are *more complex* than the real-valued parameters or binary decisions that are often encountered in standard practice. Yet there is nothing preventing statistical inference from being applied to such complex objects. Our paper makes this case and shows one way to operationalize it (bootstrap resampling over circuits, pairwise Jaccard as a stability metric).
> > >
> > > ---
> > >
> > > If we stopped here, we agree that the work could reasonably be called a position paper. But our contribution goes further:
> > >
> > > Acknowledging that statistical inference should be performed does not tell us *how much* it matters in practice. How damaging has it been that the MI community has skipped it, and how damaging would it be to keep skipping it? *Where* does instability actually enter the pipeline, and how severe is that instability? These are open questions specifically because the issue has received little to no attention.
> > >
> > > For instance: One might reasonably expect that edge score aggregation during circuit selection would decrease instability (small fluctuations in continuous CMA scores should sometimes flip a borderline edge but leave the circuit largely intact). But since this has not been systematically investigated, we cannot assess whether the circuits produced by current methods are trustworthy.
> > >
> > > This motivates the series of **specific empirical questions** we answer:
> > > - *How much* instability is there and where is it distributed (can we quantify it at *each stage* of the pipeline, from the continuous CMA scores to the final discrete circuit)?
> > > - How are the *findings of SOTA methods* affected?
> > > - Does the final circuit construction pipeline *increase* or *decrease* the overall variability?
> > > - What are the main *sources of instability* (data variability, hyperparameters, optimization randomness, etc.)?
> > >
> > > We designed a **large-scale empirical study** to answer these questions: ~6.45 million edge-level CMA evaluations and ~5,700 circuits generated across three models, three tasks, and multiple controlled perturbation regimes. Our results reveal several phenomena, some of which going beyond what the abstract principle that "estimates have variance" would otherwise predict:
> > >
> > > 1. **Circuit construction does not absorb score-level instability:** Bootstrap Jaccard averages only 0.561 after edge selection, meaning roughly half of the edges differ between resamples. This shows that the initial noise propagates into large structural differences in the final circuits.
> > >
> > > 2. The **instability is particularly strong:** Under exact CMA, ~70% of edge scores on Greater-Than, 97% on IOI and nearly 100% on SVA exhibit CV > 1 (see our responses to reviewers yYgF and njky). In other words, most scores change sign across inputs from the same distribution. This suggests that causal roles are input-dependent and that this is not simply a low sample size issue.
> > >
> > > 3. **Different estimators disagree fundamentally:** different EAP variants applied to the same data produce Jaccard overlap as low as 0.071 (Table 2), despite using the same underlying CMA quantities.
> > >
> > > 4. **The stable recovered structure is very small:** only 7/32,491 edges appear in over 72% of discovered circuits (see our response to njky), while the rest of what gets reported as "the circuit" is unstable.
> > >
> > > 5. Bootstrap Jaccard distributions sometimes show **bimodal structure** (Figure 3), which is consistent with non-identifiability.
> > >
> > > Our practical recommendations follow from these observed failure modes rather than simply general principles. For example, we propose reporting bootstrap Jaccard (n $\ge$ 100) with a tentative threshold of 0.8 (see our response to njky, yYgF). We calibrated this threshold against our finding that most circuits currently do *not* meet this threshold. We also recommend sweeping intervention strength, based on our discovery of a critical regime (amplitude $\approx 0.2$) where instability spikes. This would be invisible without a systematic empirical study.
> > >
> > > In summary, what distinguishes our work from a position paper is the fact that we identify **specific open empirical questions** about the reliability of MI practice, answer them using **controlled large-scale experimentation**, and propose **actionable guidance tied to concrete failure modes**.
> > >
> > > We propose restructuring the introduction to explicitly highlight these empirical questions, making it clear that our contribution is not merely observing that MI practice should be better, but also measuring **where, how, and by how much** it falls short.

---

### Decision · Program_Chairs · 2026-04-30

**Decision:**

Accept (regular)

**Comment:**

This paper treats circuit discovery in mechanistic interpretability (MI) as a statistical estimation problem, decomposing the pipeline into stages (per-input CMA scoring, gradient approximation via EAP, dataset aggregation, threshold-based selection) and measuring instability at each. Through ~6.45M edge-level CMA evaluations and ~5,700 circuits across three tasks and three models, the authors show instability is large, propagates through the pipeline, and yields structurally divergent circuits.

Scores: 5 (`njky`, confidence 4), 5 (`5z4f`, confidence 4, raised from 4), 4 (`yYgF`, confidence 3, raised from 3), 4 (`RBdC`, confidence 4). Mean 4.5. `RBdC` adjusted their score to a weak accept during discussion after engaging with the authors' responses. The central debate was whether these findings go beyond reminding the MI community of standard statistical principles. `RBdC` initially argued the observations are expected from basic reasoning, but ultimately stated: "I am convinced by the authors response that this level of systematically quantified illustration of the impact of the overlooked standard statistical practice in Mech Interp is a useful and important reminder for the community." `5z4f` maintained that "attributing overall instability to the variance of exact CMA, instead of attributing it to approximation error, is not obvious a priori."

The contribution is diagnostic, not methodological — no new method or estimator is proposed. What tips it over the acceptance threshold is that the quantitative results are specific enough to be useful: bootstrap Jaccard averaging 0.561 after edge selection, only 7/32,491 edges in >72% of circuits, estimator disagreement as low as 0.071 Jaccard. These numbers inform how reported circuits should be interpreted, and the CMA-vs-approximation attribution was not established prior to this study.

Scope limitation — exact CMA experiments are feasible only on gpt2-small. The camera-ready should be precise about this, as `yYgF` noted the instability "should not be claimed as universal or fundamental, but still meaningful for moderate-sized models."